# Random-Access Infinite Context Length
# for Transformers

**Amirkeivan Mohtashami**
EPFL
amirkeivan.mohtashami@epfl.ch

**Martin Jaggi**
EPFL
martin.jaggi@epfl.ch

## Abstract

While Transformers have shown remarkable success in natural language processing, their attention mechanism's large memory requirements have limited their ability to handle longer contexts. Prior approaches, such as recurrent memory or retrieval-based augmentation, have either compromised the random-access flexibility of attention (i.e., the capability to select any token in the entire context) or relied on separate mechanisms for relevant context retrieval, which may not be compatible with the model's attention. In this paper, we present a novel approach that allows access to the complete context while retaining random-access flexibility, closely resembling running attention on the entire context. Our method uses a landmark token to represent each block of the input and trains the attention to use it for selecting relevant blocks, enabling retrieval of blocks directly through the attention mechanism instead of by relying on a separate mechanism. Our approach seamlessly integrates with specialized data structures and the system's memory hierarchy, enabling processing of arbitrarily long context lengths. We demonstrate that our method can obtain comparable performance with Transformer-XL while significantly reducing the number of retrieved tokens in each step. Finally, we show that fine-tuning LLaMA 7B with our method successfully extends its context length capacity to over 32k tokens, allowing for inference at the context lengths of GPT-4. We release the implementation of landmark attention and the code to reproduce our experiments at https://github.com/epfml/landmark-attention/.

## 1 Introduction

Large transformers have revolutionized language modeling and demonstrated remarkable abilities to perform various tasks with zero or few examples [4]. This success can be largely attributed to the attention mechanism, which allows each token to access the representation of any other token in each layer. However, this flexibility comes with quadratic computational cost and highly problematic memory footprint, limiting the number of tokens that can be attended to, and thus the context length.

To overcome this limitation, researchers have proposed various solutions, including incorporating a form of recurrent memory inside the Transformer architecture, such as Transformer-XL [9]. However, these approaches often sacrifice the random-access flexibility of attention.

An alternative approach to overcome the context length limit is to use retrieval-based methods that incorporate additional static knowledge by searching for relevant documents in a knowledge base and adding them to the context. However, this approach requires a separate mechanism to identify relevant documents, called a retriever. Such retrieval models can not easily be updated to work on fresh long input data, and furthermore are also not fully compatible with the standard attention mechanism itself, and thus may fail to mimic attention over long documents.

In this work, we propose a novel approach for overcoming the context length limit by allowing earlier blocks of the input to be directly incorporated into the attention itself. We break the input into blocks

37th Conference on Neural Information Processing Systems (NeurIPS 2023).

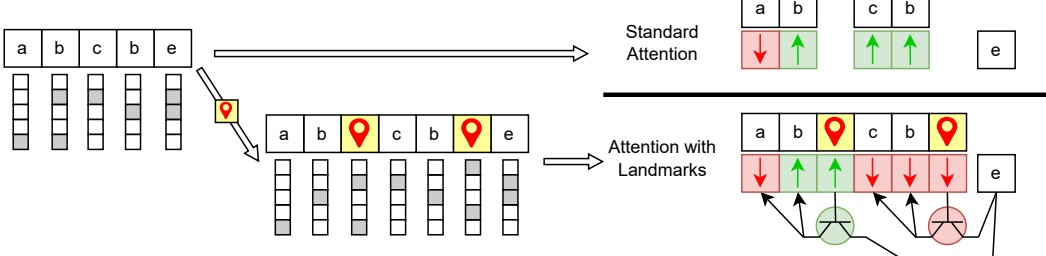

Figure 1: An illustration comparing *standard attention* and our *attention with landmarks*. The example shows the (causal) attention given by a current token e to previous ones, illustrating our mechanism with block-size $\ell_{\text{block}} = 2$. The attention scores rely on the similarity of query vector with the key vector, and in our case also with the landmark vector corresponding to the block. This is why the same token b can have a high (green) attention score when being part of one block and a low (red) attention score when being in other one, despite having the same representative vector in both cases. Landmark tokens (same as regular tokens) have the same vector representation at the first layer. However, this changes as they are updated though depth, leading to the illustrated behavior of attention at the intermediate layers.

of fixed length and introduce a special token for each block, called a landmark, which acts as a gate for attending to its corresponding block. The gating mechanism is controlled by the attention score to the landmark token. At inference time, the attention scores on the landmarks allow us to retrieve any previous block and integrate it with standard attention. The idea is illustrated in Figure 1. Our proposed approach maintains the random-access flexibility of attention and offers an alternative solution to the recurrent memory approaches.

Our model can process any context length at inference time regardless of the context length used at training time. To achieve this, we split the input into chunks and feed the chunks sequentially to the model, while maintaining a cache of previous tokens, usually referred to as KV cache. When processing each chunk, we first use the landmark tokens to select the most relevant blocks and only use these blocks for computing the attention. This immediately reduces the computation cost by a factor of block length. For example, in our experiments where we use blocks of 50 tokens, this translates to almost 50x reduction of computation. We note that to the overhead of computing the attention for the retrieved blocks does not depend on the input length and becomes negligible for very large inputs. Furthermore, it is possible to obtain the same reduction in memory usage since all tokens in a block (except the landmark itself) can be swapped out and only loaded when the corresponding landmark token is activated (see Appendix G). We also point out that this reduction can be further improved by using special data structures designed for retrieving closest neighbors, such as FAISS [15].

We demonstrate the efficacy of our method in practice by applying our training method both for training models from scratch and for fine-tuning pre-trained models. In both cases, our model effectively utilizes landmark tokens to retrieve relevant blocks from memory, enabling inference at arbitrary context lengths much longer than those encountered during training. As a result, our model obtains comparable performance with Transformer XL trained to use recurrence on a much larger window. More importantly, we demonstrate that using our method to fine-tune LLaMA 7B [38], a large language model, allows it to retrieve relevant information from contexts with over 32k tokens, which is the context length of GPT-4 [25].

The primary advantages of our method can be summarized as follows:

- Enabling inference at any context length, irrespective of the context length utilized during training, without incurring additional training costs.

- Instantly reducing inference time and memory usage (compared to a model trained to operate at the given context length) by a substantial factor equal to the block size (e.g., 50 in our experiments).

- Compatibility with advanced data structures that can further decrease the resource requirements for operating the model with very large context lengths.

## 2    Related Work

With the evolution of state-of-the-art commercial models and their applications towards very long context window lengths, such as 32k tokens (GPT-4 [25]) or even 100k (Claude [2]), the research question of efficient while accurate long context models is receiving increased attention.

**Retrieval-Augmented Language Models.**    Retrieval-augmented language models use a separate module, called a retriever, to find a set of relevant documents in the knowledge base, which are then prepended to the input. The augmented input is then fed into the main model, called the reader. Various methods have been proposed for training retrievers and readers [16]. For example, REALM [11] jointly trains the reader and retriever, where both components are transformers. Atlas [13] further investigates the effect of various losses on the performance of the retriever. Previous work has also looked into using the attention in the reader to build a retriever but used manually crafted rules to reduce the token scores to document scores [14, 18, 31]. In contrast, our landmark approach is trained to directly produce meaningful landmark embeddings on the fly, without needing any notion of corpus for retrieval.

**Memory for Transformers.**    Various methods have been proposed to introduce memorization capabilities to Transformers through recurrence [5, 40]. Transformer-XL [9] feeds the input to the model in windows of a fixed length and allows each token to attend to tokens in the current window as well as the preceding window. Memory Transformers [6] introduce special memory tokens that are prepended to the input, and their representation at the final layer of the model is used for the next input. Infinite Memory Transformers [23] map the input to a continuous space and then sample points to be used for memory in the next step according to the probability distribution defined by the attention mechanism. However, while these methods improve upon the memory-less variants, they do not allow for attending to *specific* tokens in the past, as the model only has access to a compressed version of this information. In fact, Mu et al. [24] in simultaneous work propose adding special "gist" tokens which are trained to summarize the prompt so far, and find that the model is incapable of remembering specific details that should be copied into the output. Furthermore, the decision about whether to keep or discard a piece of information needs to be made without knowledge of future tokens, which makes it likely that aspects of information will be lost, especially if the topic changes. In contrast, using our method, the model always has the possibility of retrieving and attending to any tokens in the past. Nonetheless, we note that these methods can be combined with ours, allowing the model to benefit from both full access to previous tokens as well as access to a summarized version in terms of the recurrent memory state.

**Approximate and Sparse Attention.**    Various methods have also been proposed to reduce the memory footprint of attention. However, similar to recurrent memories, these approximations significantly reduce the flexibility of attention in attending to arbitrary individual tokens. For example, Child et al. [7] limit the attention to a local window around each token, while BigBird additionally suggests attending to a random subset of previous tokens as well as several globally accessible tokens [42]. Longformer [3] further introduces dilated sliding window patterns to increase attention's receptive field and manually picks the window sizes for each layer. Linformer [39] uses a low-rank approximation of the attention matrix while Performer [8] uses a non-softmax kernel to obtain a more efficient implementation. Reformer [19] uses locality-sensitive-hashing (LSH) to retrieve the closest key vectors which should account for the highest scores in the attention matrix. Combiner [30] utilizes a hierarchical attention mechanism and heuristic reduction techniques, such as max-pooling, to derive key and query vectors for input blocks. The block weight is determined based on the pooled key vector, while the weight of each token within the block is determined by the pooled query vector. However, this approach limits the control of the current token over the weights of the tokens inside the block, resulting in reduced flexibility of attention. In contrast, our proposed method enables the current token's query vector to control the weight for each token, and the gating mechanism is learned through the attention process instead of relying on heuristic reductions.

$k$**NN Augmented Transformers.**    $k$-nearest-neighbor ($k$NN) augmentation has been proposed as an alternative method for allowing transformers to access external memory. For example, $k$NN-LM [17] stores the hidden representation of tokens in memory and uses the distribution of the next token among the stored vectors that are closest to the current token to predict the next token. Memorizing Transformer [41] performs a nearest-neighbor search over previous keys and computes the attention to the top nearest items. However, these methods obtain the final results by interpolating between the $k$NN prediction and the local attention prediction using a tuned parameter as interpolation weight. Therefore, the interpolation does not depend on the current input and does not consider whether the memory contains relevant information.

**Context Length Extrapolation.**    Transformers have a well-known limitation in extrapolating to contexts longer than what was observed during training [27], even when relative positional encoding is used [36]. Current solutions to address this problem often result in weakened attention scores for long-range tokens, which undermines the benefits of a longer context [27, 36]. Moreover, these methods only work when combined with windowed attention, which restricts direct attention to long-range tokens [36]. We hypothesize that the limitation may partially stem from the model's learning of its own positional encoding with the use of causal masking, as demonstrated in [12]. This limitation poses a challenge as our goal is to enable access to long-range tokens during inference at distances that were not observed during training. We discuss solutions in Section 3.2.1.

## 3    Methodology

In this paper, we mainly focus on the causal language modeling where each token can only attend to previous tokens in the input. We briefly discuss the extension of our method to the non-causal case in Appendix F.

When using a Transformer to process a long input, the ideal case would be to allow each token to attend to all previous tokens. However, this becomes computationally infeasible as the input length increases. Nevertheless, since the attention scores always sum to one, the number of keys with a large attention weight is limited even for long contexts. Thus, by retrieving only those keys with large attention scores, it is possible to closely emulate the ideal case. In this work, we propose a method to find these keys by dividing a long input into blocks of consecutive tokens and using the attention to retrieve relevant blocks.

More particularly, we assign a representative vector to each block such that a high attention score to any token inside a block would lead to a high attention score to the block's representative vector. Therefore, we can directly retrieve blocks based on the attention score of their representative vector.

To obtain the representative vector of a block, we introduce a new special token to the vocabulary, called the landmark token. We insert a landmark token after the last token of each block and train the model such that the key vector for this token becomes the representative vector we seek. The process is illustrated in Figure 1.
We will first describe the method we use to train the landmark tokens in Section 3.1 and then describe the inference process in Section 3.2.

We note that an alternative for directly finding a candidate set of keys with high attention score is using a data structure that allows finding nearest neighbors of the query vectors efficiently such as FAISS [15]. In comparison, our method provides a retrieval method directly controlled by attention which can be more semantic-based. Furthermore, retrieving a block instead of a single token allows the attention to also access the local context around the token which may be more accommodating of observed classic attention patterns [20]. Finally, we point out the aforementioned data structures can also be applied on top of our method to search for relevant blocks.

### 3.1    Training Landmark Tokens

In order to train the landmark tokens, we first go over the text corpus and add a landmark token after every $\ell_{\text{block}}$ tokens. Then we proceed to training the model using the standard batching method which feeds windows of $\ell_{\text{seq}}$ tokens to the model. In a list of $\ell_{\text{seq}} > \ell_{\text{block}}$ tokens, we train the model such that each landmark token represents the block consisting of all previous tokens until the previous landmark token (or the beginning of the input if no previous landmark token exists). The token is passed through the transformer as any other token while its representation is updated using the self-attention mechanism. Let us denote the index (token position) of the landmark corresponding to the $i$-th token's block by $p_i$. If the last block is incomplete and does not have a landmark token, we define $p_i := \ell_{\text{seq}}$. If the $i$-th token is a landmark token, $p_i := i$ .

In order to train the transformer to make use of landmark tokens, we alter the standard attention mechanism such that the attention weight for a token depends on the similarity of the query vector with both the token's key as well as with the key of its block's landmark token. To define the mechanism, we first define a generalized softmax function called Grouped Softmax. Given a vector $\mathbf{v} \in \mathbb{R}^{\ell_{\text{seq}}}$ and a group index $\mathbf{g} \in \mathbb{N}^{\ell_{\text{seq}}}$, Grouped Softmax applies softmax separately over elements

belonging to the same group. (Using $\mathbf{g} = \mathbb{1}_{\ell_{\text{seq}}}$ recovers the standard softmax function):

$$\sigma_G(\mathbf{v}, \mathbf{g})_x := \text{GroupedSoftmax}(\mathbf{v}, \mathbf{g})_x := \frac{e^{\mathbf{v}_x}}{\sum_{y : \mathbf{g}_y = \mathbf{g}_x} e^{\mathbf{v}_y}} \ . \tag{1}$$

We replace the softmax function after computing the attention scores with Grouped Softmax. For each block, we put its regular tokens in a separate group, ensuring that all regular tokens within the same block share the same group, while tokens outside the block are assigned to different groups. When computing the attention weights for the $i$-th token, landmark tokens for other blocks are placed in the same group as the $i$-th token. The landmark token for the $i$-token's block is **ignored** when computing the attention weights for the $i$-th token. In other words, the landmark token for each block is only used by tokens in other blocks. This is intuitive as the landmark token should only be accessed when tokens in other blocks require to retrieve information from the landmark's corresponding block. Building on the fact that $p_j = j$ only holds when the $j$-th token is a landmark token, we can define the grouping used for the $i$-th token more formally as

$$\mathbf{G}_{i,j} := \begin{cases} p_j & p_j \neq j & \triangleright \text{ placing normal tokens in their own blocks.} \\ -1 & p_i = j & \triangleright \text{ ignoring current block's landmark token.} \\ p_i & p_i \neq j \wedge p_j = j & \triangleright \text{ placing other landmarks in the } i\text{-th token's group.} \end{cases} \tag{2}$$

Finally, to obtain the final weights after applying GroupedSoftmax, we multiply each token's softmax output with the softmax output for its block's landmark token. For the tokens in the same group as the $i$-th token, we directly use the softmax output as its attention weight. The weight for landmark tokens is always zero. In more formal terms,

$$\mathbf{S}_{i,j} := \text{SoftmaxScore}(\mathbf{Q}, \mathbf{K})_i \quad := \quad \text{GroupedSoftmax}\left( \frac{\mathbf{Q}_i^\top \times \mathbf{K}}{\sqrt{d_{\text{head}}}}, \mathbf{G}_i \right) \tag{3}$$

$$\text{AttWeight}(\mathbf{Q}, \mathbf{K})_{i,j} \quad := \quad \begin{cases} 0 & p_j = j \\ \mathbf{S}_{i,j} & \mathbf{G}_{i,j} = \mathbf{G}_{i,i} \wedge p_j \neq j \\ \mathbf{S}_{i,j} \cdot \mathbf{S}_{i,p_j} & \mathbf{G}_{i,j} \neq \mathbf{G}_{i,i} \wedge p_j \neq j \end{cases} \ . \tag{4}$$

An exmaple illustration of various values defined above is given in Appendix A. Note that under this scheme, the attention weights sum to one as is the case for the standard softmax function. More importantly, attending to tokens in other blocks is gated by the attention score to the landmark token as expected. Since tokens in the same block and the landmark tokens share the softmax group, the model has to choose between attending to other blocks and current tokens. Thus, the intuition behind the grouping is to force the model to only attend to relevant blocks due to this trade-off.

We note that attention masks can be applied normally by ignoring the masked elements in the softmax (e.g. by setting $A_{i,j}$ to $-\infty$ on the masked elements in practice). Indeed we focus our experiments on the causal language modeling. We also point out that the grouping scheme can be further extended to introduce additional hierarchy for retrieval. For example, we refer the interested reader to Appendix D, where we briefly discuss a different grouping scheme which also trains a global retrieval gate token that controls whether retrieval from an earlier block needs to be performed. At inference, this gate can be used to decide whether a memory call is needed or the model already has the information it needs in the context. We leave further investigations of this setting for future work.

### 3.2 Inference

Similar to training, the input gets augmented by a landmark token after every $\ell_{\text{block}}$ tokens. Then, we break the input into chunks of $\ell_{\text{local}}$ length and iteratively feed chunks from the beginning to the end. To retrieve relevant blocks, each attention layer has access to a cache of previous blocks. The cache stores the key-value vectors for all tokens of those blocks, including the landmark token. Since the retrieval only requires access to the landmarks, we can reduce the memory usage significantly by swapping out (for example to CPU memory or even to disk) all regular tokens' cached key-value vectors, and then swapping them back in only if their corresponding block is retrieved by the attention.

We start by discussing the most permissive retrieval scheme. When processing each chunk at each layer, we first compute the attention score of each token with the landmark tokens currently in the cache. For each token, we compute the attention score for all the tokens in **the $k$ highest scoring blocks** and prepend the obtained attention matrix to the local attention matrix. We finish by applying

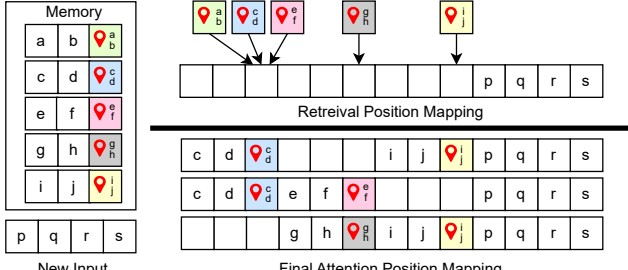

Figure 2: Stingy position mapping: Retrieving top $k=2$ blocks from a memory of 5 blocks. Retrieval landmarks for the last 2 blocks are accurately mapped to sequence index positions, while previous blocks are mapped to the position of the $(k+1)$-th last block. Blocks are then distributed across the prefix based on their position, with an empty block separating the last $k$ blocks from older ones.

$\mathrm{GroupedSoftmax}$ to the attention matrix and computing the weighted average of the value vectors to obtain the token's representation.

Under the above scheme, each token and each head can retrieve different blocks from the cache. It is possible to limit the retrieval flexibility in order to improve efficiency. For example, it is possible to merge the scores across heads by taking a maximum over the landmark attention scores (after applying softmax) of each head. Under this scheme, the same set of blocks is retrieved for all heads. It is also possible to take the maximum over different tokens, retrieving only $k$ blocks per head for all the tokens in the current window combined. We study the effect of these limitations at the end of Section 4.1. Unless otherwise stated, we experiment using the permissive scheme described above.

### 3.2.1 Positional Encoding

When computing the attention scores to cache elements (both landmark and normal tokens), it is important to correctly incorporate positional information. The transformer model is sensitive to positional information. It is also intuitive that the model would rely on position information in some cases. For example tokens right after the last memory block do not have access to any context and are unable to select memory blocks based on semantics. Instead, they need to rely on positional information to select the last memory block.

Optimally, the tokens are encoded with their actual position index. However, a known flaw of Transformers is their inability to extrapolate to lengths not observed during training. Various methods proposed to alleviate this condition also do not fully resolve the problem unless they are combined with block attention which only allows attending to a window of tokens. We decide against using block attention since our main goal is to facilitate attending to large contexts. In Appendix E we propose an alteration of the training method which allows using the actual position index. However, to avoid overlapping changes, we use the following approximation for most of our experiments.

We allocate a segment with length $(k+1) \cdot (\ell_{\text{block}}+1)$ in the beginning of the context. We index the current chunk starting after this segment. For the retrieved blocks, we map the index for any of the latest $k$ blocks to the corresponding place within the last $k$ blocks in the allocated prefix segment. Other blocks in the cache have their position mapped to the first block in the allocated prefix segment. Once we decide which blocks to retrieve, we map those among the latest $k$ blocks to the right of pre-allocated segment and map the rest of the blocks to the left of the pre-allocated segment, while respecting the order of the blocks. We call this scheme *stingy position mapping* which is further illustrated in Figure 2.

Note that we found out that when $k=1$ mapping memory blocks to a segment of at least 2 blocks is crucial. Using only a single block, all memory blocks are mapped to the same index. However, as we discussed at the beginning, it is essential to at least retrieve the last memory block solely based on position information which is impossible unless this block is mapped to a different index. While it is possible that the importance of pre-allocating the additional block decreases as $k$ grows, we adapt this scheme so that the attention would be at least as flexible of simply keeping the last $k$ blocks.

We point out that the above approximation relies on the ability to add position information when performing the retrieval. In our experiments, we use Transformer models with Rotary positional encoding [33] which adds the position information to the key and query vectors just before computing

the attention. Thus, we can store the key vectors without position information in the cache and add the position information when performing the retrieval according to the following scheme.

### 3.3 Memory & Computation

During training, our method has only a negligible overhead due to the computation of GroupedSoftmax. In particular, our method does not require maintaining a cache of previous values at training time. Furthermore, we decouple the training context length from the inference context length since it is possible to perform inference at any context length using the method described in Section 3.2 regardless of the train context length. As such, when comparing training time in terms of inference context length, we offer constant training time ($\mathcal{O}(1)$) whereas training time for a standard transformer scales quadratically with the operational (inference) context length.

Furthermore, in comparison with standard attention over the whole input, our method reduces inference time by computing the attention score over a smaller set of token pairs. For instance, during auto-regressive generation, where tokens are generated sequentially by using the previously generated token as input for obtaining the next one, the traditional approach involves computing attention across all preceding tokens when generating the next token. In contrast, our method allows computing attention solely over landmark tokens, followed by attention computation over the retrieved blocks. Given the constant size of the blocks, the cost of computing the attention over the retrieved blocks remains constant regardless of the total context length. While the cost of finding the most relevant landmark tokens increases linearly with the context length, the rate of increase is 1 every $\ell_{\text{block}} + 1$ tokens. This immediately reduces the number of operations by a factor of block length $\ell_{\text{block}}$. For example, when using $\ell_{\text{block}} = 50$ as in our experiments, this can lead to a 50x boost. Importantly, the same reduction can be obtained in terms of memory. In the standard Transformer, a cache of all previous key and values (KV-cache) is needed to perform the generation efficiently. In contrast, we only need immediate access to the landmark tokens and can offload the blocks to slow memory (e.g. CPU), loading only the retrieved blocks. The reduction in memory and compute can be further improved if the search for retrieving blocks is performed using more advanced data structures such as FAISS [15].

It is worth noting that the additional computational overhead introduced by performing two matrix multiplications (one for block selection and another for attention to the retrieved blocks) instead of a single matrix multiplication in the standard setting becomes relatively negligible, especially when dealing with larger inputs.

Finally, we point out that our method can be naturally combined with flash attention [10], reducing the overhead further. We discuss this in Appendix F. In this work, to reduce the complexity and allow flexibility in experiments, we use a high-level implementation (not combined with Flash Attention). However, we also publish a version of efficient implementation in Triton [37].

## 4 Experiments

### 4.1 Language Modeling

We first evaluate the efficacy of retrieving earlier blocks on two language modeling tasks which can be expected to have long-range token interactions: English language books (PG-19) [29] (3.7B tokens), and math papers from arXiv (5.6B tokens). We provide additional details about the datasets in Appendix B. Our results show that models trained with landmark tokens can retrieve relevant blocks, obtaining comparable perplexity as a Transformer-XL while reducing FLOPs. In contrast with Transformer-XL, using our method, the information retrieval is interpretable since the exact tokens attended to by the model can be identified by looking at the attention scores or looking at the set of retrieved blocks. Particularly, it is possible to understand which parts of the text was recovered to generate a certain answer, which can for example be useful to remove inaccurate information. Our results also demonstrate that using the inference mechanism described in Section 3.2, our models can be used at much longer context than the one used for training.

**Model & Training.** We use a GPT-2 [28]-like architecture: a 12-layer decoder-only transformer with 8 heads in each layer, each with dimension 128 (embedding dimension 1024), and hidden FFN size of 4096. We trained our model using AdamW [22] with $\beta_1 = 0.9$ and $\beta_2 = 0.95$. We applied weight decay with factor 0.001. We used base learning rate 0.002 for all our experiments with a warmup stage that was 2% of the whole training and applied a cosine scheduler with minimum (final)

Table 1: Performance of different training and inference settings in terms of language modeling perplexity. The column XL cache shows the size of the XL cache available both during training and inference which was only used when training Transformer-XL[9]. When using landmarks, the column "⚑ Blocks" shows the maximum number of blocks stored in memory. Each block contains $\ell_{block} = 50$ normal tokens and one landmark token. Due to computation limitations we only report results for Transformer-XL on PG-19 as this method takes longer to train in our implementation.

| Eval. Length | $\ell_{local}$ | XL cache | ⚑ Blocks | $k$ | Attention Size | PG19 | arXiv | |
|---|---|---|---|---|---|---|---|---|
| 512 | 512 | None | None | - | 512 | 16.12 | 4.01 | Baseline |
| | 360 | None | None | - | 360 | 16.76 | 4.31 | |
| | 250 | None | 10 | 2 | 360 | 16.23 | 4.01 | Ours |
| 2048 | 256 | 256 | None | - | 512 | 14.72 | - | [9] |
| | 250 | None | 40 | 2 | 360 | 15.14 | 3.43 | |
| | 350 | None | 40 | 2 | 460 | 15.07 | 3.41 | Ours |
| | 300 | None | 40 | 3 | 460 | 14.97 | 3.36 | |
| | 250 | None | 20 | 4 | 460 | 15.02 | 3.37 | |
| | 250 | None | 40 | 4 | 460 | 14.92 | 3.35 | |
| 4096 | 256 | 256 | None | - | 512 | 14.55 | - | [9] |
| | 250 | None | 40 | 4 | 460 | 14.79 | 3.19 | |
| | 250 | None | 80 | 2 | 370 | 15.00 | 3.29 | Ours |
| | 250 | None | 80 | 4 | 470 | 14.72 | 3.18 | |

learning rate being 0.0004. We used GPT-2's [28] tokenizer. When using landmark tokens, the tokens were added to the dataset and stored as part of the train dataset, leaving the batching mechanism unchanged. We used gradient accumulation as well as data-parallel training across four nodes to maintain an effective total batch size of 128. We used mixed-precision training with `bfloat16` over at most 4 Nvidia A100 GPUs. For our method, we train the model on each dataset for 240K steps with context length $\ell_{seq} = 512$. We train Transformer-XL with a window size of 256 (i.e. effective context size 512) over segments of length 2048. We train Transformer-XL to observe the same number of tokens during training as our method which translates to performing 60K steps.

**Results.** To evaluate our model's performance with different context lengths, we divide the validation data into equally sized segments, referred to as evaluation lengths. Each segment is separately inputted into our model, which is further divided into chunks using the method described in Section 3.2. The chunk size, denoted as $\ell_{local}$, represents the local context accessible without any memory. Table 1 presents the perplexity of the trained models under various inference settings. Notably, by using a local context length of 250 and retrieving the top $k = 2$ most relevant blocks, we achieve a comparable performance with a context length of 512. This corresponds to attending to 360 tokens, including 250 tokens from the local context, 10 landmark tokens, and 100 tokens from the retrieved blocks. The effectiveness of using landmark tokens with retrieval becomes even more evident when comparing it to standard inference with an attention length of 360. Our results demonstrate that intelligently recovering relevant blocks enables attending to a significantly smaller number of tokens while maintaining performance.

Furthermore, our results highlight that landmark tokens enable the model to operate with larger context lengths than those encountered during training. The improvement in perplexity clearly indicates that the retrieved blocks contribute to the model's performance, making the results comparable to a Transformer-XL trained with segments of length 2048. However, unlike Transformer-XL, which can only leverage past information through recurrence, our method allows the model to attend to any token from the past, facilitating both the retention of exact fine-grained details and the interpretability of information utilization mechanisms.

Finally, the number of retrieved blocks and the number of blocks stored in memory can be adjusted during inference. While reducing the number of retrieved blocks $k$ adversely affects performance, our results demonstrate that the model still outperforms the baseline even with only 2 retrieved blocks at context lengths of 2048 and 4096. Notably, when keeping only the last 40 blocks in memory, the model performs better at an evaluation length of 4096 compared to 2048. This suggests that the model is also learning to utilize recurrent mechanisms similar to those in Transformer-XL.

Table 2: Performance on PG19 dataset for different levels of retrieval flexibility. The blocks column shows the theoretical total number of blocks that can be accessed from the memory when feeding the input in windows of length 250 to the model.

| Per Head | Per Token | Eval. Length | $k$ | Blocks | Perplexity |
|:---:|:---:|:---:|:---:|:---:|:---:|
| | | 2048 | 2 | $250 \cdot 8 \cdot 2$ | 15.14 |
| ✓ | ✓ | 2048 | 4 | $250 \cdot 8 \cdot 4$ | 14.92 |
| | | 4096 | 4 | $250 \cdot 8 \cdot 4$ | 14.72 |
| | | 2048 | 2 | $8 \cdot 2$ | 15.48 |
| ✓ | ✗ | 2048 | 4 | $8 \cdot 4$ | 15.10 |
| | | 4096 | 4 | $8 \cdot 4$ | 14.95 |
| | | 2048 | 2 | $250 \cdot 2$ | 15.44 |
| ✗ | ✓ | 2048 | 4 | $250 \cdot 4$ | 15.04 |
| | | 4096 | 4 | $250 \cdot 4$ | 14.89 |

**Granularity of Cache Block Retrieval.** Block retrieval can be performed on different levels of granularity. At the most granular level, the set of retrieved blocks can be different for each head and each token. This setting is the same as the model experiences during training. However, it is possible to further limit this granularity at inference, for increased system throughput. In this section we evaluate the effect of maintaining the same set of retrieved blocks across tokens or across heads. The results are presented in Table 2 which also shows the total number of retrieved block, with the same block retrieved by different token or head counted multiple times. While reducing the flexibility has a noticeable adverse effect on performance, the model still improves over the baseline. In particular, we note that it is possible to retrieve the same set of blocks for all tokens (which varies across heads) while only suffering 0.23 points in perplexity. To provide further insights into the expected improvement in speed gained from using a less flexible selection scheme, we further discuss the distribution of the retrieved blocks in Appendix C.

## 4.2 Fine-Tuning Pre-Trained Models

We demonstrate the possibility of fine-tuning a large language model using landmark's token and therefore extending the model's context length. Namely, we fine-tune LLaMA 7B [38] for 15000 steps using our method. To reduce computation, we fine-tune the model with context length 512. We use the sample subset of RedPajama[1] for the fine-tuning which closely follows the dataset curation process used for training LLaMA.

We evaluate the efficacy of our method by comparing model's ability to recover a hidden pass phrase inside a text segment. In particular, we use randomly generated prompts of the format described in Figure 3a and compute the accuracy of generating the correct pass key (as the first integer within the first 100 generated tokens). The result is plotted in Figure 3b for different context lengths. We observe that the base model is capable of finding the pass phrase until a certain lengths, even slightly exceeding its default training context length of 2048 (the area shared in grey). However, the base model completely fails at the task for larger contexts. In contrast, our landmark version can always retrieve the pass phrase with high accuracy, even for significantly larger context lengths. We point out that when evaluating our model with very large inputs (e.g. 32K), we use additional techniques to reduce the memory usage by offloading the KV cache (execpt the landmarks) to CPU. We discuss this in more detail in Appendix G.

## 5 Future Work

**Extrapolating Positional Encoding.** One of the obstacles in attaining infinite context length is the inability of models to attend to context lengths much larger than those they were trained on. In this work, we provide a special indexing method which can be combined with landmark tokens to bypass this issue. However, as a result, the model can only attend to tokens that are too far based on their semantic (and not their position). While this is an important improvement and facilitates extrapolation to large context lengths, it can be expected that the performance would be further improved if the exact indexing method can be used. Unfortunately, existing proposals limit (or completely disable)

---

[1]https://github.com/togethercomputer/RedPajama-Data

```
There is an important info hidden inside
a lot of irrelevant text. Find it and
memorize them. I will quiz you about
the important information there.
<prefix filler by continuously repeating:
The grass is green. The sky is blue.
The sun is yellow. Here we go. There
and back again.>
The pass key is <PASS KEY>. Remember it.
<PASS KEY> is the pass key.
<suffix filler>
What is the pass key? The pass key is
```

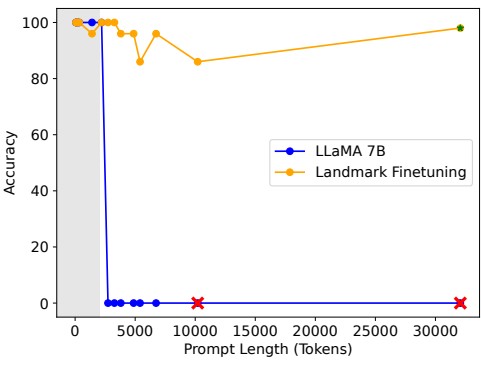

(a) Prompt Format                    (b) Retrieval Accuracy

Figure 3: Prompt format used for comparing retrieval accuracy of the vanilla LLaMA 7B and its counterpart fine-tuned with landmarks. The points marked with a red cross represent cases where the model ran out of memory. Points marked with a green star use a more efficient inference mechanism (see Section G). Inference is done by feeding the segment in windows of length 250 tokens (excluding the inserted landmark tokens). The top $k=4$ landmarked blocks are retrieved. Retrieval accuracy is measured for a fixed total prompt length, by using the suffix and prefix filler. Results are averaged over 50 random generation of the pass key (a random number between 1 and 50000), which each time is located at a random position in the full-length prompt. The space before and after the pass key is filled accordingly by the suffix and prefix filler. The gray box marks the region where the prompt length is within lengths used during original LLaMA training.

attention to far tokens which defeats our purpose. While we briefly discuss a possible solution for models with landmark tokens in Appendix E, we leave a more thorough investigation as future work. We note that once such method is developed, it can be directly combined with landmark tokens, yielding inference capabilities at any length.

**Hierarchical Landmarks.** In large-scale settings, the landmark tokens can be stored in k-nearest neighbor data structures to improve retrieval performance and reduce memory usage. However, an alternative is to introduce hierarchy with higher level landmark tokens controlling the attention to lower level landmarks. In Appendix D, we investigate adding a special token which acts as a gate to all landmark tokens. This token can for example be used to decide whether a retrieval is necessary. Similarly, this token can be used at different memory cache levels where high attention to this token would constitute a cache miss, leading to lookup in lower-level (and slower) caches. We leave exploration of possible hierarchical landmark tokens as a future direction.

**Training with Cache.** For simplicity, in this work we focus on using the standard training procedure. While we expect the standard softmax mechanism to closely resemble the retrieval at inference, given the special indexing scheme, it is possible that the model would gain additional benefit from incorporating the cache during training. We leave investigation of such training variants as a future work.

## 6    Conclusion

In conclusion, this work presents a novel method for training attention to retrieve relevant blocks from memory. Unlike previous methods that rely on recurrence to create memory, our approach enables direct access to previous tokens, ensuring accurate information retrieval without the problem of slowly forgetting past data. We have demonstrated that our method achieves comparable performance to recurrent methods such as Transformer-XL while utilizing less computational resources. Additionally, our attention-based retrieval process allows for tracking and interpretability, providing insights into the information used to generate the output. Importantly, our results highlight the ability of our approach to handle significantly longer context lengths than those encountered during training. Moreover, we have shown that this capability can efficiently be incorporated into existing pre-trained models through fine-tuning, showcasing improved retrieval capabilities in the LLaMA 7B language model. Overall, our method enables efficient inference with arbitrary context lengths, making it suitable for accessing large inputs and processing fine-grained information within the large context.

## Acknowledgment

The authors would like to express their gratitude to Matteo Pagliardini for insightful discussions and Olivia Simin Fan for their invaluable feedback on initial drafts of this paper. This project was supported by SNSF grant number 200020_200342.

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

## A   Grouped Softmax Example

| $i$ | 0 | 1 | 2 | 3 | 4 | 5 | 6 | 7 | 8 |
|---|---|---|---|---|---|---|---|---|---|
| $\text{mask} + \dfrac{Q_6^T K}{\sqrt{d_{\text{head}}}}$ | 1 | 1 | 1 | 1 | 1 | 1 | 1 | mask | mask |
| $p_i$ | 2 | 2 | 2 | 5 | 5 | 5 | 8 | 8 | 8 |
| $G_{6,j}$ | 2 | 2 | 8 | 3 | 3 | 8 | 8 | 8 | -1 |
| $S_{6,j}$ | 0.5 | 0.5 | 0.33 | 0.5 | 0.5 | 0.33 | 0.33 | 0 | ign |
| $W_{6,j}$ | 0.167 | 0.167 | 0 | 0.167 | 0.167 | 0 | 0.33 | 0 | 0 |

Figure 4: Example illustration of applying the method used for applying attention through landmark tokens. In the above example, a causal mask is applied as well. Tokens at indices 2, 5, and 8 are landmarks in the example as can be seen by checking the condition $p_i = i$. The example is based on the assumption that the dot product of the query vector of the token at index 6 and other key vectors is an all-one vector. It can be seen that when computing the attention for this token, the landmark at position 8 is ignored while the other two landmark token, act as gate for attending to tokens in their block. The attention weight to landmark tokens is distributed among normal token in their block. Therefore, the final attention weight to the landmark tokens themselves is always zero.

## B   Dataset Description

**arXiv Math**   We use the cleaned arXiv math subset of proof-pile [2] dataset. The dataset contains all the TEX files for submissions under math category in arXiv. Files with encodings other than `utf-8/16/32` or `latin1` were removed. Additionally, the dataset curation includes certain heuristic to only keep well-structured papers. An example is removing files that do not contain any chapter, sections, or subsections. The final training dataset contains around 5.6B tokens.

**PG-19**   PG-19 is a large dataset (3.7B tokens in the training dataset) of English books that were published before 1919 and were retrieved from the Project Gutenberg archive. This dataset is widely used for evaluating model capabilities in utilizing long-range token interactions [35, 41].

## C   Number of Unique Retrieved Blocks

We feed the validation set of PG19 in segments of at length 2048 (in chunks of 250 tokens). During this process, we retrieved the top $k = 4$ blocks from the cache and kept track of the number of unique blocks retrieved in the penultimate chunk. Note that we count the same block retrieved by two different tokens only once. To ensure consistent statistics, we utilize the penultimate chunk instead of the last one, as the last chunk may have a lower number of tokens that could disrupt the analysis. We record this number separately for each batch element and each layer and plot the distribution at different retrieval flexibility levels in Figure 5. In the penultimate chunk, the cache contained 35 blocks. As depicted in the figure, at the most flexible level, it can be observed that in many cases, all the blocks are accessed at least once at each layer, as indicated by the spike in the last bin. Interestingly, the additional spike in the flexible retrieval corresponds to a very low number of blocks

---

[2]https://github.com/zhangir-azerbayev/proof-pile/

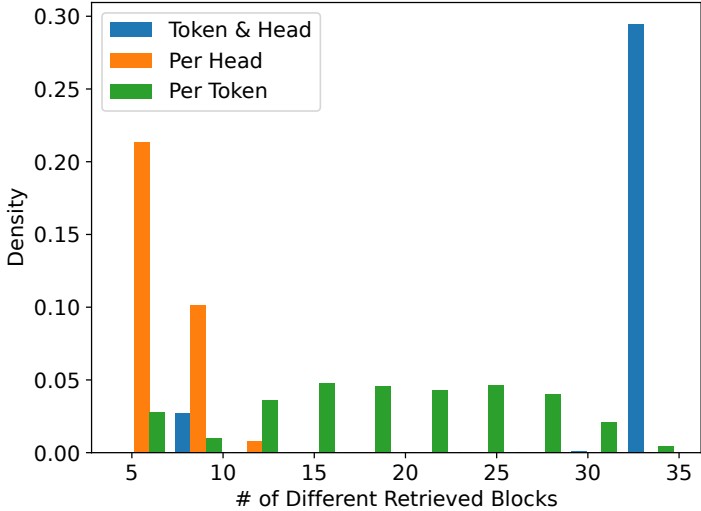

Figure 5: The distribution of the number of unique retrieved blocks by the 255 tokens (including the landmark tokens) in the penultimate chunk when evaluating over 2048 token inputs from PG19.

being retrieved and is attributed to the retrieval pattern in the initial layer. This finding aligns with previous research suggesting a locality of attention in the earlier layers [34].

Moreover, the results demonstrate a significant decrease in the number of unique retrieved blocks when only allowing the set of retrieved blocks to vary across heads. Although there are 8 heads, each retrieving 4 blocks from the cache, theoretically allowing access to 32 different blocks, we typically observe the number of unique blocks to be below 10. While this reduction does come at the cost of perplexity (as observed in Table 2), it can significantly improve performance by facilitating offloading to slower devices through a reduction in the required bandwidth for loading retrieved blocks from the cache. It is worth noting that one approach to mitigate the negative impact of reduced flexibility on perplexity is to increase the number of retrieved blocks. This can be observed by comparing the results between retrieving the top $k = 2$ blocks at the highest level of flexibility and retrieving the top $k = 4$ blocks with reduced flexibility, as shown in Table 2. By retrieving more blocks, we can partially compensate for the reduced flexibility and its effect on perplexity.

Finally, when allowing the set of retrieved blocks to vary across tokens but not heads, it is still possible to observe improvement in the number of unique retrieved blocks which can reduce the bandwidth needed to load blocks from the cache. At the same time, better perplexities can be achieved in this setting.

## D  Context Miss Token

In this section, we demonstrate how to create additional hierarchy in the landmark structure by simply changing the grouping. In particular, we use a new grouping scheme to train a new special token, called context miss token (CMT). CMT is always placed at the beginning of the input (at the special position $-1$) and is used to signal the need to retrieve landmarked blocks from the memory. To train this token, we change the grouping scheme so that attention to some of the blocks is regulated by CMT, similar to how landmark tokens act as gateways to the tokens in their respective blocks. In particular, let us denote $L_{\text{CMT}}$ as the set of landmark tokens that are controlled by CMT. Then, using the same notation as in Section 3.1, we use the following grouping during training (For brevity, in the below formulation we assume that the negative of all preceding conditions hold for each case):

$$\mathbf{G}_{i,j} := \begin{cases} p_i & j = -1 & \triangleright \text{ placing CMT in the } i\text{-th token's group.} \\ p_j & p_j \neq j & \triangleright \text{ placing normal tokens in their own blocks.} \\ -1 & p_i = j & \triangleright \text{ ignoring current block's landmark token.} \\ p_i & j \notin L_{\text{CMT}} & \triangleright \text{ placing free landmarks in the } i\text{-th token's group.} \\ -2 & j \in L_{\text{CMT}} & \triangleright \text{ placing CMT-controlled landmarks in a separate group.} \end{cases} \quad . \quad (5)$$

Using the above grouping the attention weights can be calculated as:

$$\text{AttWeight}(\mathbf{Q}, \mathbf{K})_{i,j} := \begin{cases} 0 & p_j = j \vee j = -1 \\ \mathbf{S}_{i,j} & \mathbf{G}_{i,j} = \mathbf{G}_{i,i} \\ \mathbf{S}_{i,j} \cdot \mathbf{S}_{i,p_j} & j \notin L_{\text{CMT}} \\ \mathbf{S}_{i,j} \cdot \mathbf{S}_{i,p_j} \cdot \mathbf{S}_{i,-1} & j \in L_{\text{CMT}} \end{cases} \quad . \quad (6)$$

To build intuition, we note that using an empty $L_{\text{CMT}}$, almost recovers the original formulation (with the exception of the existence of CMT token which can absorb part of the attention score). Note that assuming $L_{\text{CMT}}$ is not empty the above formulation ensures that the total sum of attention weights is equal to one. During training, we suggest to choose $L_{\text{CMT}}$ randomly by selecting each landmark token with probability $\mathbb{P}_{\text{CMT}} = 0.5$.

Intuitively, the above formulation should teach the network to attend to the CMT token whenever it does not have enough information within the free landmark tokens (landmark tokens that are not controlled by CMT). Furthermore, we note that since CMT is the first token, it can not attend to any token. As such, its representation in each layer does not depend on the input. As such, the token acts as a beacon for signaling the context lacks enough information. That is why we draw a parallel to the notion of a cache miss, naming it a context miss token. We note that CMT can be used to create different levels of landmark block storage, going to a lower (more granular) level whenever it is activated at each level, similar to a cache miss signal. We leave a thorough investigation of such designs to future work.

To evaluate our method, we follow the configuration used in Section 4.1 and train a model with CMT on PG19. At inference we set $L_{\text{CMT}}$ to be the set of landmarks that are in the memory so the in-context landmarks are not controlled by the CMT. Using this setting, the CMT can be used to check whether a retrieval is necessary or not. For simplicity, in our implementation we always perform the retrieval but set $\mathbf{S}_{i,-1}$ to 0 if it is below a cutoff threshold. This exactly emulates not performing the retrieval at all. We report the perplexity on PG19 with different cutoff thresholds and the ratio of CMT scores below the cutoff threshold that were subsequently dropped in Table 3. Our results show that it is possible to use CMT to significantly reduce the number of retrieval calls while still utilizing the memory. In particular, around 50% of the retrievals can be dropped with minor effect on perplexity. Note that the difference observed between training without CMT (baseline) and training with CMT (no cutoff) is reasonable since the model is learning a harder task. We conjecture that this difference can be alleviated by simply training the model longer.

## E  Positional Augmentation

In our main method, we use a special position mapping scheme (See Section 3.2.1), called Stingy mapping. The need for Stingy mapping is due to the model's inability to extrapolate to positions that were not observed during training [36]. Existing methods allow the model to handle longer inputs by manually dampening or completely limiting attention to far-away tokens [27, 36]. These techniques are not applicable to our settings since one of our main goals is to allow attention to any token, including those that are very far. While development of a positional encoding scheme which can extrapolate is outside the scope of this work, we propose a possible solution and provide results of an early investigation. We leave a more thorough assessment and development of such schemes for future work.

Data augmentation has been used in various settings to allow the model to generalize to additional settings such as reflections of the same image [21]. We propose to apply augmentation on positional

Table 3: Performance on PG19 dataset with 2048 evaluation length and $k = 4$ for different CMT cutoff thresholds. When the GroupedSoftmax for CMT is below the cutoff threshold, it is set to zero to emulate not performing a retrieval. The drop rate column shows the ratio of CMT scores below the cutoff threshold. Baseline refers to the model with landmarks but without CMT. The models are trained for 60K steps.

| Cutoff | Perplexity | Drop Rate |
|---|---|---|
| Baseline | 16.28 | 0% |
| 0.0 | 16.38 | 0% |
| 0.1 | 16.38 | 23% |
| 0.3 | 16.43 | 57% |
| 0.5 | 16.86 | 84% |
| 1.0 | 19.49 | 100% |

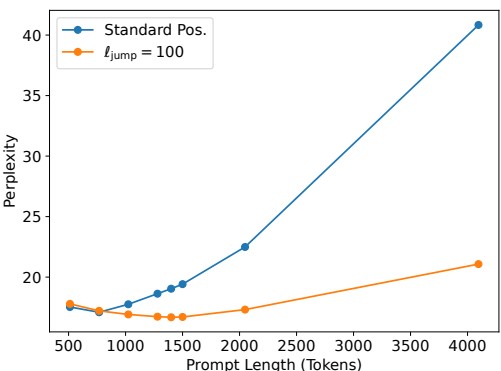

Figure 6: Comparison of perplexity on PG19 for different context lengths. The evaluation does not use the landmark cache and feeds the whole input in a single iteration to the model. Still, the landmark tokens are inserted every 50 tokens. The context length does not include the additionally inserted landmark tokens.

information in Transformers to allow them to extrapolate to longer contexts. In the standard positional encoding, the positions are increased by one at each token, leading to the tokens being assigned positions 1 to $\ell_{\text{seq}}$ where $T$ is the length of the input. In particular, instead of assigning positions from 1 to $T$, where $T$ is the length of the input, we propose to increase the positions of all subsequent tokens by a random integer between 1 and $p_{\text{jump}}$ after each landmark token. We refer to these increases as making positional jumps. When $p_{\text{jump}} = 1$, no augmentation is applied and the standard positions are recovered.

To evaluate our proposal, we train the same model that we used in Section 4.1 on PG19 with $p_{\text{jump}} = 100$. Since we use a context length of 512 for training, each input can has between 10 and 11 landmark tokens. Theoretically, this should allow the model to extrapolate to context lengths as long as $1100 + 512 = 1612$. We plot the performance of the model on different context lengths as well as the performance of the model with standard positional encoding (i.e. $p_{\text{jump}} = 1$) after 60K steps in Figure 6. Note that we do not use a cache in this section and feed the whole input at once.

We can see that using the augmentation, the model becomes capable of utilizing longer contexts. This is evident by the fact that we observe reduction in perplexity as we increase context lengths until 1400 tokens which is close to the theoretical estimate of model's extrapolation capacity. In contrast, the decreasing trend stops for the standard model before reaching 1024 tokens. We can also observe that when evaluating at training context lengths, i.e. 512 tokens, the performance when making positional jumps is lower than standard training. However, this can be expected and justified since the model is learning a harder task and therefore may require additional steps to reach the same performance.

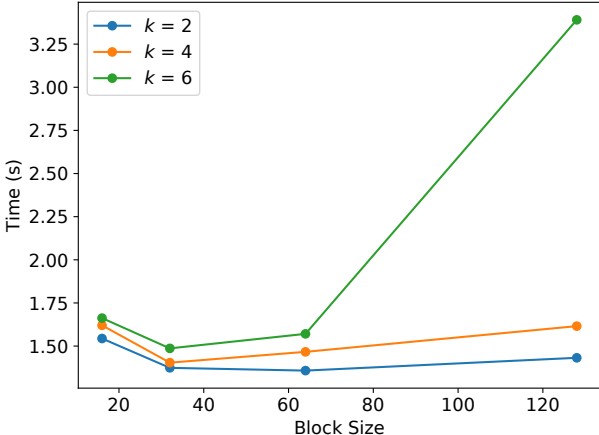

Figure 7: Time needed to process $4870$ tokens (excluding the landmarks) using Landmark Attention for different number of retrieved blocks $k$ and different block sizes. The efficient implementation of landmark attention combined with FlashAttention is used. The retrieval flexibility is reduced so that all tokens in each chunk use the same set of retrieved blocks.

## F  Additional Extensions and Details

**Masked Language Modeling**    In this work, we focus on next-word prediction tasks. In this section, we briefly discuss how to adapt our scheme for masked language modeling. Our discussion here is conceptual, and experimental verification of our proposal is left for future work. We note that the scheme described in Section 3.1 can be used to train a model with masked language modeling. However, certain changes need to be made to the inference method. Specifically, the input cannot be given to the model sequentially from left to right because each token should be able to access the subsequent tokens as well. Thus, we propose processing the entire input layer by layer. In each layer, the input is broken into chunks, and each chunk is processed separately, with each token retrieving the top-$k$ landmarked blocks from other chunks.

**Combination with Flash Attention**    Flash attention computes the attention matrix in blocks [10] with the block size chosen to optimize memory accesses. Landmark Attention can be naturally combined with flash attention by choosing the frequency of the landmark tokens to be equal to the aforementioned block size. Note that in this case, the calculation of the GroupedSoftmax incurs negligible overhead (both in terms of memory and compute) because only the softmax values for the currently processed block and the landmark block need to be kept and adjusted at each step. Combination with more advanced extensions is also possible. For example, the landmark token can naturally be used to decide on dropping blocks when using block sparse flash attention. Finally, when reducing the retrieval flexibility and enforcing the same set of retrieved blocks for all tokens in the chunk, Flash Attention can be directly applied at inference as well. We implement this combination using Triton and show that it can be used to fine-tune LLaMA7B with 2048 context length (instead of the maximum 512 tokens possible without flash attention).

**Trade-off between number of retrieved blocks and block size**    During inference, when using landmark attention, landmark tokens need to be also stored in memory which slightly increases the memory usage and compute time by a factor of $\frac{1}{\text{block size}}$. Thus increasing block size reduces memory usage while also reducing the time needed to find relevant blocks since there are less blocks. However, if the number of retrieved blocks $k$ is kept constant, the length of the chunks have to become smaller so the longer retrieved blocks and the current chunk can fit into the model's maximum allowed context length, i.e. the train context length. Therefore, increasing the block size increases the number of chunks the input has to be broken into, which slows down the inference. This trade-off can be seen in the Figure 7, showing the inference time for different block sizes and different values of $k$.

.

# G  Offloading KV Cache to CPU

Previous research has proposed various techniques to reduce the size of the KV cache [1, 26, 32], which can be directly applied to address this bottleneck. However, using landmark attention allows to reduce memory usage further using a simpler approach: When using landmarks, it is possible to offload the cache to the CPU and only load back the retrieved blocks in GPU at each step, while keeping all landmarks in GPU memory. While for inference with very high context lengths the techniques proposed by previous research would still be necessary, offloading to CPU allows us to perform inference with LLaMA 7B with more than 32K tokens.

Although the aforementioned technique works well, it introduces significant CPU-GPU traffic, resulting in slow inference. To mitigate this issue, we limit the number of retrieved blocks by reducing retrieval flexibility, allowing the set of retrieved blocks to vary across heads but not across tokens. We use the same reduction mechanism as in Section 4.1. Specifically, we calculate the maximum score for each block across all tokens and select the top $k = 5$ blocks for retrieval. Prior to selecting the maximum, softmax is applied to ensure all scores are on the same scale.

Using the above method, we evaluated the fine-tuned LLaMA with a context length of 32070 tokens (the exact number of tokens may vary slightly due to random generation, but the difference is less than 10 tokens). Across 50 randomly generated prompts, we achieved an encouraging accuracy of 98% in retrieving the pass key. This clearly demonstrates the model's capability to effectively operate at context lengths comparable to those supported by GPT-4 [25].

