# OpenReview forum: "Random-Access Infinite Context Length for Transformers"
_NeurIPS.cc/2023/Conference — NeurIPS 2023 poster_

### Official Review · Reviewer_tGXa · 2023-06-16

**Soundness:** 2 fair
**Presentation:** 3 good
**Contribution:** 3 good
**Rating:** 5
**Confidence:** 4

**Summary:**

The paper presents a new architecture for long-range decoder-transformers: a new "landmark token" is inserted into the input within constant strides (that is, after every $k$ tokens). Every such "landmark token" is thus the "representative" of a block of tokens.
The attention score to this "landmark token" is treated as a multiplicative gate for attending the tokens within its block.

That is, during training: tokens in the local proximity are attended to as usual; the attention score of far away tokens is their usual token-attention probability, multiplied by their landmark's attention probability.

During inference, the model attends to the local proximity tokens (the last k tokens) and to the previous landmarks, chooses the top-k landmarks, and attends only to the local block's tokens (the most recent tokens) and to tokens within these top-k blocks.
In other words, the model always attends to the most recent tokens, but performs a kind of "hierarchical attention" to the past tokens.

This allows, after fine-tuning, extending LLaMa 7B from 2k to 10k inputs.

**Strengths:**

## Strengths
* The approach allows finetuning the newest models (such as LLaMA) to extend their input length significantly
* The discussion of related work is thorough, albeit with some inaccuracies regarding Memorizing Transformers (see Weaknesses below).


**Weaknesses:**

## Weaknesses
- The authors claim "infinite context length", but demonstrate only on ~10k tokens. The authors write that "it is theoretically possible for the model to access any token in the entire past" - but this was not demonstrated. Even if the authors argue that this is only a matter of missing engineering, it needs to be demonstrated that the model can actually leverage the attention to the entire past in an effective way to claim "infinite context length" (although I believe that even solely the engineering is not as simple as the authors try to make it sound).
- The paper notes that a model that was trained this way can be used as a general-purpose document retriever, but this was not demonstrated as well.
- The authors make some claims about the most related work, [Memorizing Transformers (Wu et al., ICLR 2022)](https://arxiv.org/pdf/2203.08913.pdf), that are inaccurate:
> Memory Transformers [33] ... However, while these methods improve upon the memory-less variants, they do not allow for attending to *specific* tokens in the past, as the model only has access to a compressed version of this information

This is incorrect: Memorizing Transformers [33] do allow for attending to specific tokens in the past, by indexing and retrieving them in a kNN index. The main difference may be that in Memorizing Transformers, the trained gate is applied to the entire attention head, rather than to the local "landmark". Thus, I also think that the claim about having access only to a compressed version of the information is incorrect - later in the Related Work, the paper says:
> Memorizing Transformers [33] performs... However, these methods obtain the final results by interpolating between the kNN prediction and the local attention prediction using a tuned parameter as interpolation weight

This is again incorrect. Memorizing Transformers does not perform interpolation with a tuned parameter, it attends to the retrieved kNN tokens with a learned attention, as in standard transformers (the claim in the paper is correct regarding kNN-LM, but not to Memorizing Transformers)

* The explanation about positional encoding (Section 3.2.1) is very unclear. What does "allocate a segment" mean? where is this allocated and why? What does "*we index the current chunk starting after this segment*" mean? I could not parse sentences such as "*For the retrieved blocks, we map the index for any of the latest k blocks to the corresponding place within the last k blocks in the allocated prefix segment*", although I am very familiar with the long-range transformers literature and with Rotary positional embeddings.
* Evaluation
    * Table 1 shows the language modeling evaluation.
        * What metric is this? The table only shows numbers without mentioning the metric. is this perplexity?
        * From what I am able to understand, in the longest evaluation length of 4096, Transformer-XL [7] performs better than the proposed approach? (achieving perplexity of 14.55 on PG19 which is lower than "Ours").
        * In evaluation length of 2048, Transformer-XL [7] also performs better than "Ours", achieving perplexity of 14.72.
        * Am I right to conclude that the proposed approach outperforms only the vanilla base transformer, but not outperforming Transformer-XL?
        * The authors argue that "*In contrast with Transformer-XL, using our method, the information retrieval is interpretable. Particularly, it is possible to understand which parts of the text was recovered to generate a certain answer*" - however this interpretability was not demonstrated, so I did not take it as an advantage.
    * The experiments with LLaMA-7B were demonstrated only on a synthetic task
    * baselines:
        * The only baseline is Transformer-XL, which came out in 2019.
        * [Zhong et al., "Training Language Models with Memory Augmentation", EMNLP '2022](https://arxiv.org/pdf/2205.12674.pdf) discusses several retrieval scenarios - retrieving from an external datastore, and using kNN retrieval for long inputs. I believe that their TRIME-long is a very relevant baseline. I agree that TRIME-long is conceptually more limited because it uses a tuned interpolation hyperparameter as in kNN-LM, but the insight there of training with retrieval *from other examples in the same batch* may compensate for it.

**Questions:**

## Questions
1. What is the metric in Table 1? perplexity?
2. Am I right to conclude that the proposed approach outperforms only the vanilla base transformer, but not outperforming Transformer-XL?
3. The paper clearly discusses relevant long-range transformers in the Related Work section. Isn't any of them a relevant baseline?
4. The only baseline is Transformer-XL, which came out in 2019. Can Memorizing Transformers and TRIME$_{\mathrm{long}}$ be relevant baselines?
5. In the LLaMa experiments in Section 4.2, the paper mentions that LLaMA was fine-tuned with a context length of 512 tokens. For fairness, was the proposed model trained with a context of 512 as well?

## Additional Comments
6. Possibly related paper [Bertsch et al., "Unlimiformer: Long-Range Transformers with
Unlimited Length Input"](https://arxiv.org/pdf/2305.01625.pdf). I refer to Unlimiformer as concurrent work since it appeared on arxiv 2 weeks before the NeurIPS deadline, so a direct comparison is not required.
7. I think that the explanation of GroupedSoftmax and the entire Section 3.1 can be simplified, as it explains the operator from the implementation point of view, rather than from the conceptual idea. I think that it could have been explained more simply by just saying that "every token attends only to the tokens within the same block, and to the other landmarks of other blocks", without bothering the reader with the $\mathbf{g}$ indices. Papers such as [Longformer](https://arxiv.org/pdf/2004.05150.pdf) also implement attention with masking and summing over indices, but the explanation in the paper is much more intuitive (see Figure 2 in [Longformer](https://arxiv.org/pdf/2004.05150.pdf)).
5. Equation (2) uses $\mathbf{G}$ rather than $\mathbf{g}$, unless $\mathbf{G}$ is something else; in that case, $\mathbf{G}$ was not defined.

## Summary
I like the overall approach and I think that it can be very useful in practice. However, there are a few significant issues:
1. Overclaiming, and claiming abilities that were not demonstrated, but only "theoretically possible"
2. Lack of baselines - the only baseline is Transformer-XL, which performs better (!) than the proposed approach

I am thus voting for only a borderline acceptance at this time. I would increase my score if the above questions, issues and weaknesses would be resolved.

---

> ### Author Rebuttal · Authors · 2023-08-08
>
> Dear Reviewer tGXa,
>
> We make the following comments:
>
> 1. Our argument on infinite context length relies on the model's capabilities. While practical demonstration of infinite context length isn't feasible, our method's efficacy is shown in Appendix G at 32k context length. The original transformer remains crippled by the inability of positional encodings to fully generalize to arbitrary length. Current solutions sacrifice direct attention to early tokens to address this. However, our approach, using stingy positional mapping, eliminates this and reduces quadratic complexity by the block size factor.  While inevitably going to larger lengths requires more resources, we show that this can be further alleviated by offloading the cache to CPU which is what we use in Appendix G. Our experiments reveal the model's ability in key retrieval despite stingy positional mapping, justifying the term "infinite." We welcome any suggestions to make our argument more concise.
>
> 2. We only note the ability to use for retrieval as a future prospect. We will clarify this in the final revision.
>
> 3. Memory Transformer is different from Memorizing Transformer. By mistake, we have referenced the wrong paper in Line 79. We will fix this in the final revision and cite Burtsev et. al. (2020). Indeed Memory Transformers do not allow for attending to specific tokens in the past while Memorizng Transformers do.
>
> 4. We refer to Section 3.1 of the Memorizing Transformer paper where the interpolation is discussed. The knn-augmented layer in memorizing transformers similar to kNN-LM performs an interpolation between the memory and the LM output to obtain the final output.
>
> 5. Please note that we still use standard Rotary Positional Encoding in our model. Our approach solely modifies token indexing (not the encoding) to eliminate the need for extrapolation to indices unseen at training. Here is an alternative description: We feed the input to the model in chunks; as in the paper. So the total number of tokens in the chunk and in the retrieved blocks should be less than or equal to the training context length $l_{\text{train}}$. So we introduce $l_{\text{train}}$ placeholders corresponding to indices 1 to $l_{\text{train}}$ to be filled with tokens from either the chunk or retrieved blocks, as shown in Figure 2. By allocating a segment in the beginning of the input we mean reserving some of the early placeholders. The placeholders beyond the reserved segment are filled with the chunk tokens left to right. Note that the token to placeholder (index) allocation differs between finding relevant blocks and attending to them. When attending to relevant blocks, those that are among the last $k$ blocks fed to the model are right-aligned in the reserved segment while others are left-aligned. Given that the reserved segment spans $k + 1$ blocks, this leaves one empty block between the two groups. We refer to Figure 2 for a visual representation of this process.
>
> 6. As specified in Line 282, the numbers in Table 1 are perplexity scores. We will make it clear in the caption of the Table in the final revision.
>
> 7. We present Transformer XL as a baseline to show that landmark attention can perform comparably in utilizing long contexts. However, Transformer XL has inherent limitations as it cannot directly access earlier tokens, hindering its ability to perform certain tasks, such as retrieval. In contrast, our LLaMA experiments demonstrate that Landmark Attention successfully retrieves and attends to early tokens.
>
> 8. Please note that the information in Transformer XL has to be passed through recurrence which prevents identifying the tokens the model attends to. In this sense, our model is more interpretable since the exact tokens attended to by the model can be identified by looking at the attention scores or looking at the set of retrieved blocks, i.e. blocks with highest scoring landmarks.  We will clarify this further in the final revision.
>
> 9. We want to strongly emphasize that **in our method each token can attend to any other token**. In our method, a token attends to specific tokens in other blocks as well. However, the attention scores to those tokens are gated by their block's landmark token's score (through multiplication as in Eq 4). We provide high level intuition for the method in the introduction and Fig 1 but formally define our method in Sec. 3.
>
> 10. We do not train a separate model in LLaMA experiments and only compare a fine-tuning of LLaMA using our method at length 512 with Meta's vanilla LLaMA (not fine-tuned) at length 2048, our baseline. If we have misunderstood your question, please let us know.
>
> 11. We emphasize a key distinction between Memorizing transformers and our approach: Memorizing transformers require training with a large context length. This increases training cost or complicates implementation due to the need for linking the model to FAISS data structure. Due to the increased training cost it is challenging to train such models for large contexts. In contrast, our method allows inference at arbitrary context lengths regardless of the training context length. We tried training our model using the TRIME-Long method. Unfortunately, the training collapsed after 140k steps. At that point, Landmark Attention noticeably outperforms TRIME on PG-19 dataset in addition to being cheaper to train. The perplexity with 4096 tokens is 15.54 for landmarks and 17.43 for TRIME. We also note that the Trime-Long method is inherently limited since the model can not directly attend to the past context and intuitively it can only match close repetitions.
>
> 12. $g$ is any vector and denotes one of the inputs of GroupedSoftmax in Equation 1. $G$ is the specific grouping matrix defined in Equation 2.
>
> We hope that the above replies adequately addresses your concerns and that you would consider raising your score. Please also read our general reply. We remain at your disposal to answer any additional questions or comments.

---

> > ### Comment · Reviewer_tGXa · 2023-08-14
> > **Response to authors**
> >
> > Thank you for your response.
> >
> > >1. Our argument on infinite context length relies on the model's capabilities...
> >
> > I still think that demonstrating empirically 10k tokens (or 32k in a synthetic dataset) and calling it "infinite context" is still a bit of a stretch.
> >
> > >2. We only note the ability to use for retrieval as a future prospect. We will clarify this in the final revision.
> >
> > I suggest moving this future prospect to the end of the paper. When reading such claims at the beginning of the paper, the reader expects to see empirical evidence of them.
> >
> > >3. Memory Transformer is different from Memorizing Transformer. By mistake, we have referenced the wrong paper in Line 79.
> >
> > OK, it's great that we caught it. So the Wu et al. 2022 Memoriz**ing** Transformers is not discussed? What are the conceptual advantages and disadvantages of this work compared to Memoriz**ing** Transformers?
> >
> > >4. We refer to Section 3.1 of the Memorizing Transformer paper where the interpolation is discussed. The knn-augmented layer in memorizing transformers similar to kNN-LM performs an interpolation between the memory and the LM output to obtain the final output.
> >
> > Right, but this paper says that it does that "using a **tuned parameter** as interpolation weight" which is incorrect, Memorizing Transformers **learn** this interpolation weight, so I don't see that learning this is necessarily a bad thing (and the paper mentions that as a weakness as far as I understand).
> >
> > >5. Here is an alternative description: We feed the input to the model in chunks; as in the paper. So the total number of tokens in the chunk
> > > ...By allocating a segment in the beginning of the input we mean reserving some of the early placeholders...
> > > The placeholders beyond the reserved segment are filled with the chunk tokens left to right. Note that the token to placeholder (index) allocation differs between finding relevant blocks and attending to them. When attending to relevant blocks, those that are among the last
> >  blocks fed to the model are right-aligned in the reserved segment while others are left-aligned. Given that the reserved segment spans
> >  blocks, this leaves one empty block between the two groups...
> >
> > I still think that this can be explained more intuitively. Maybe avoiding the words "allocation" and "segment" and focusing on the mathematical idea (rather than the implementation) would make it clearer.
> >
> > >7. We present Transformer XL as a baseline to show that landmark attention can perform comparably in utilizing long contexts. However, Transformer XL has inherent limitations as it cannot directly access earlier tokens, hindering its ability to perform certain tasks, such as retrieval. In contrast, our LLaMA experiments demonstrate that Landmark Attention successfully retrieves and attends to early tokens.
> >
> > To justify this claim, I would expect a non-synthetic experiment that shows this. What if access to early tokens is not necessary in any realistic task, so in practice Transformer XL should always be used instead of the proposed model?
> >
> >
> > >8. lease note that the information in Transformer XL has to be passed through recurrence which prevents identifying the tokens the model attends to. In this sense, our model is more interpretable since the exact tokens attended to by the model can be identified by looking at the attention scores or looking at the set of retrieved blocks
> >
> > These claims about being "more interpretable" sound reasonable in theory, but they were not demonstrated, and I thus cannot seriously consider them as an advantage against Transformer XL. Transformers in general are "somewhat interpretable" (but largely not), so comparing their relative interpretability without any demonstration is a bit of a stretch.
> >
> > >9. We want to strongly emphasize that in our method each token can attend to any other token. In our method, a token attends to specific tokens in other blocks as well.
> >
> > Can't Memorizing Transformers attend to any other token as well?
> >
> > >10. We do not train a separate model in LLaMA experiments and only compare a fine-tuning of LLaMA using our method at length 512 with Meta's vanilla LLaMA (not fine-tuned) at length 2048, our baseline
> >
> > The LLaMA model that was fine-tuned with the proposed approach and length 512 - on which data was it fine-tuned?
> >
> > >11. Memorizing transformers require training with a large context length. This increases training cost or complicates implementation due to the need for linking the model to FAISS data structure.
> >
> > I agree that it seems that it's easier to train the proposed approach compared to Memorizing Transformers. However, Memorizing Transformers can theoretically be also trained with only a memory size that fits in the GPU memory, without FAISS at all. I think that the authors have conducted such experiments when the memory is limited during training.
> >
> > Best,
> > Reviewer `tGXa`

---

> > > ### Author Response · Authors · 2023-08-15
> > >
> > > Dear Reviewer tGXa,
> > >
> > > Thank you for considering our replies and providing further comments. We hope the following replies address your questions.
> > >
> > >  > OK, it’s great that we caught it. So the Wu et al. 2022 Memorizing Transformers is not discussed? What are the conceptual advantages and disadvantages of this work compared to Memorizing Transformers?
> > >
> > > We discuss both of these works in our paper. Memorizing Transformer is discussed in Line 114 while Memory Transformer is discussed in Line 79. The only typo is the index of referenced paper in line 79.
> > >
> > >  > Right, but this paper says that it does that “using a tuned parameter as interpolation weight” which is incorrect, Memorizing Transformers learn this interpolation weight, so I don’t see that learning this is necessarily a bad thing (and the paper mentions that as a weakness as far as I understand).
> > >
> > > The process of learning is commonly referred to as tuning the model (e.g. fine-tuning), which is why we refer to the weight as a tuned parameter (as opposed to a hyper-parameter). The problem with this approach is that the weight does not depend on the input and is completely fixed during inference. Therefore even if the memory does not contain any relevant information, the model will still put a weight on the kNN results (and vice-versa).
> > >
> > >  > To justify this claim, I would expect a non-synthetic experiment that shows this. What if access to early tokens is not necessary in any realistic task, so in practice Transformer XL should always be used instead of the proposed model?
> > >
> > > The need to copy words from early into the text can commonly arise in text generation. For example, consider the name of character that was referred to in a book several chapters ago. It is very important that the model generates the correct character name but this is only needed once after possibly many other words. Therefore, observing such improvement is not easy using metrics such as perplexity on normal text dataset. This is why we demonstrate the capability on the synthetic dataset.
> > >
> > >  > Can’t Memorizing Transformers attend to any other token as well?
> > >
> > > This is correct. We made the quoted comment only to avoid a misunderstanding given your previous summary (number 7 in your review) “every token attends only to the tokens within the same block, and to the other landmarks of other blocks”. We have provided a comparison with Memorizing Transformer both in our related works (line 114) and in our reply (number 11).
> > >
> > >  > The LLaMA model that was fine-tuned with the proposed approach and length 512 - on which data was it fine-tuned?
> > >
> > > The model was fine-tuned on RedPajama dataset which has been created with the aim to reproduce the original training data from Meta. Please note that the fine-tuning is not done for a particular task (e.g. instruction tuning, etc.).
> > >
> > >  > I agree that it seems that it’s easier to train the proposed approach compared to Memorizing Transformers. However, Memorizing Transformers can theoretically be also trained with only a memory size that fits in the GPU memory, without FAISS at all. I think that the authors have conducted such experiments when the memory is limited during training.
> > >
> > > We could not find results in the Memorizing Transformer paper suggesting that training can be done with a small memory block while using a larger memory block at inference. Intuitively using very small memory sizes at training and larger ones at inference can possibly cause problems for example with the weighting parameter (but we have not tested this).  Also please note that the bottleneck with training with a memory block without FAISS is not just RAM but also compute. For example using a memory block equal to context length doubles the computation cost. This is alleviated to some extent when using FAISS and very large memory blocks (as in the memorizing transformer paper) but still increases the step time (in some cases from 0.2s to 0.6s according to the original paper).
> > >
> > > We hope that the above replies adequately addresses your concerns and that you would consider raising your score. We remain at your disposal to answer any additional questions or comments.

---

> > > > ### Comment · Reviewer_tGXa · 2023-08-15
> > > > **thanks**
> > > >
> > > > Thank you for the clarifications.
> > > >
> > > > Just to clarify, I am not an author of Memorizing Transformers, but I think I know its details very well because I read that paper multiple times. I don't think there is a point in continue discussing the subtle details of Memorizing Transformers and whether they are advantageous or not.
> > > >
> > > > I think that the main current problem in this paper is that it over-claims a bit in many directions, without empirical justifications to back these claims:
> > > > 1. Differences and advantages compared to Memorizing Transformers (Wu et al., 2022)
> > > > 2. The "infinite-context" claim, which was demonstrated over 30k tokens in a synthetic task, where Memorizing Transformers demonstrated context of ~200k tokens.
> > > > 3. The ability to retrieve tokens from early in the text, which makes this proposed approach conceptually better than Transformer-XL, but was not evaluated in a non-synthetic task.
> > > >
> > > > Instead of the hypothetical claims of whether or not can each model have these abilities, the paper could be significantly improved if these claims were backed by meaningful experiments.
> > > >
> > > > For example, instead of such explanations:
> > > >
> > > > >The need to copy words from early into the text can commonly arise in text generation. For example, consider the name of character that was referred to in a book several chapters ago. It is very important that the model generates the correct character name but this is only needed once after possibly many other words
> > > >
> > > > It could have been much more convincing if the paper showed evaluation on book summarization or book language modeling, and demonstrated these claims empirically. Memorizing Transformers did quite a convincing job by evaluating on language modeling of source code, proofs, and arxiv papers, and showed that the model can refer to definitions that appeared 200k tokens earlier (and refers to them in practice).
> > > >
> > > > If the authors have performed such experiments, they could have demonstrated whether the proposed model really has an empirical advantage over Transformer XL. According to the experiments that are currently present in the paper, it may seem that there is no advantage over Transformer XL in the realistic tasks.
> > > >
> > > > I am still voting for acceptance, but I keep my score at borderline accept, since I think that the main claims of the paper were not demonstrated convincingly enough.

---

### Official Review · Reviewer_g7s5 · 2023-06-27

**Soundness:** 2 fair
**Presentation:** 2 fair
**Contribution:** 3 good
**Rating:** 5
**Confidence:** 5

**Summary:**

This paper presents a novel approach that enables the Transformer language models to process much longer sequences. Specifically, the authors propose to group the input sequence into multiple blocks, each of which is represented with a landmark token. The attention scores are calculated regularly among all tokens, but are normalized within each block and multiplied by the landmark scores. In this way, the attention mechanism is trained to identify and select relevant token blocks given each query. As a result, during inference, the model could dynamically select relevant context blocks for each incoming query without loading all previous contexts into memory. Combined with a careful design of positional encodings, the model is allowed to process arbitrarily long contexts and demonstrates its effectiveness in language modeling tasks across different model scales.

**Strengths:**

- The idea is novel and interesting in that it associates each contiguous block of tokens with a pointer; each incoming query is then compared against these pointer vectors (as well as tokens within the local neighborhood) and retrieves past token blocks only when the corresponding pointer is semantically relevant. A notable advantage of this approach is that the representation of these pointer vectors is learned through the attention mechanism, thereby reducing the need for heuristic reductions and enhancing the overall simplicity of the model.
- Furthermore, the exploration of positional encodings in this paper is a valuable contribution. While it is not new to investigate the impact of positional embeddings on length extrapolation, this research offers a comprehensive analysis of this phenomenon and proposes several techniques to alleviate related challenges. The empirical findings presented here not only provide insights specific to the context of this study but may also hold relevance for the general language modeling community.

**Weaknesses:**

- The paper is not that easy to follow. For example, the formulation of Equation (1) introduces confusion. The subscript $i$ in Equation (1) actually indexes the $i$-th element of vectors $v$ and $g$, rather than representing the $i$-th query as in Equation (2,3,4). I would suggest using a different subscript other than $i$ to enhance clarity. In addition, it seems that $(k+1)\cdot \ell_{\text{block}}$ in L234 should be $(k+1) \cdot (\ell_{\text{block}} + 1)$, according to the specification in L200 that “augmented by a landmark token after every $\ell_{\text{block}}$ tokens”.
- The proposed model is not evaluated sufficiently in the following aspects:
    - 1) Some baselines are missing. In my opinion, it would be beneficial to compare against other long-range methods such as Combiner (as mentioned in the paper), which processes the distant context with heuristic pooling operations.
    - 2) Ablation studies are not sufficiently extensive. For instance, this work does not examine the impact of positional encodings and block sizes in modeling long contexts;
    - 3) The conducted experiments do not provide qualitative or quantitative results supporting the claim such as “In contrast with Transformer-XL, using our method, the information retrieval is interpretable”?
- One notable advantage of this proposed model is the ability to efficiently process long contexts by enabling random access to tokens. However, this work does not demonstrate the efficiency gains (e.g., memory usage or decoding-time speedup) achieved in dealing with long contexts. Providing evidence of these gains would further strengthen the paper's contributions in this regard.

**Questions:**

1. Regarding LLAMA fine-tuning experiments, it is known that LLAMA performs poorly when it comes to length extrapolation, possibly due to positional encodings. Consequently, it remains unclear whether the improved results depicted in Figure 3(b) are solely attributed to the proposed model's enhanced ability to handle longer contexts or are primarily a consequence of the utilization of more robust positional encodings. This issue might require further investigation.
2. During inference, token blocks within the memory are frequently retrieved and replaced due to the attention landmark selection. Intuitively, such swapping-in/out processes can be computationally expensive, especially in the case of per-head and per-token selection. Does this case hold true in practical settings and are the benefits of the proposed approach only apparent in the long sequence regime?
3. How well does the model scale with longer sequences during training in terms of evaluation performance (e.g., language modeling perplexity)?

**Limitations:**

The authors have addressed the limitations; my suggestions can be found above.

---

> ### Author Rebuttal · Authors · 2023-08-08
>
> Dear Reviewer g7s5,
>
> We make the following comments to address your questions and concerns:
>
> 1. We thank you for the valuable suggestions to increase clarity (and also for pointing out the typo) and will apply them in the final revision.
>
> 2. Please note that all existing methods for handling long contexts usually require training at the target inference length, which increases the training cost, or sacrifices the ability to directly attend to any tokens, which are unlikely to perform well for retrieving an exact pass key.
>
> 3. We want to emphasize that throughout the paper we always use the standard rotary positional encoding. The stingy positional mapping is solely applied during inference and only affects how tokens are indexed, not how they are encoded for the model. Furthermore, this mapping can only be applied in combination with a method such as landmarks which identifies relevant previous blocks. It is not a separate method that can be directly used with LLaMA or other models. Therefore, the study of the effect of different positional encodings on context length is out of the scope of our work though we discuss the limitations of the existing methods at the final paragraph of the related works.
>
> 4. Overall we do not expect our method to be very sensitive to the choice of the block size. Instead, this value can be chosen based on common sense (keeping the block small enough so that retrieving a single block does not fill the context with unimportant tokens and it can be summarized effectively, but keeping it large enough to use the computational benefits). We note that the need to choose a fixed block size also can be mitigated by providing the model with blocks of different sizes at training. In early stages of our work, we briefly experimented with dropping some of the landmark tokens, thus merging consecutive blocks, and did not find a noticeable effect on performance. For simplicity we decided to avoid doing this for the final experiments.
>
> 5. Please note that the information in Transformer XL has to be passed through recurrence which prevents finding the tokens the model attends to. In contrast, in our method, the model directly attends to the tokens. In this sense, our model is more interpretable since the exact tokens attended to by the model can be identified by looking at the attention scores or looking at the set of retrieved blocks, i.e. blocks with highest scoring landmarks.  We will clarify this further in the final revision.
>
> 6. While swapping out and swapping in can lead to a slowdown, we show in Appendix G that only allowing the retrieved set to change across heads and not across tokens allows us to mitigate this problem and successfully perform inference with LLaMA at 32k inference length with 98% accuracy.
>
> 7. We acknowledge that without the reduced flexibility, i.e. in per token and per head setting, the CPU-GPU communication could become the bottleneck. However, we note that due to limitations, we did not implement caching of the retrieved blocks in GPU to avoid double communication which we expect to significantly reduce the load on CPU-GPU connection. Regardless, we emphasize that our method still improves over previous methods and allows the model to operate at arbitrary inference context lengths (though slowly) despite being trained at a much smaller context length. Therefore, using landmarks, inference at larger context lengths becomes a question of speed or having additional resources at inference time instead of the need to re-train the model.
>
> 8. In terms of perplexity, we expect our method to behave the same as the standard transformer as the context length grows. While this is not included in the paper, we can confirm for example that LLaMA 7B can also be successfully trained at 2k context length using a more efficient implementation of our method (combined with Flash Attention).  Finally, a similar method used at inference can also be applied at training to allow direct training at larger context lengths as well while reducing the quadratic computational cost by the block size factor (though we have not yet implemented this).
>
> We hope that the above replies adequately clarify our contributions and that you would consider raising your score. We would also like to ask that you read the general reply we have provided which addresses common concerns raised by the reviewers. We remain at your disposal to answer any additional questions or comments.

---

> > ### Comment · Reviewer_g7s5 · 2023-08-16
> >
> > I thank the authors for providing comprehensive and insightful responses. In general, after reading through the clarification as well as the other reviews, I decided to stay at my initial rating of 5.
> >
> > > Regarding Comments 1, 2, 3, 4, and 8:
> >
> > Thanks for the feedback. The clarification of the experimental setup makes it clear to see the flexibility introduced by landmarks, and I acknowledged that such training flexibility is advantageous in that the model does not need to be trained on target sequence length.
> >
> > > Regarding Comment 5:
> >
> > - The claims such as “improved interpretability” and “better retrieval ability” still lack empirical substantiation.
> > - If the model can attend to all tokens, it would be prudent to compare it with more relevant baselines, like sparse transformers (e.g., routing transformers) that also have this capability.
> >
> > > Regarding Comments 6 and 7:
> >
> > While I appreciate the effort made by the authors to clarify the implementation details, the empirical advantages of the proposed method in longer contexts remain ambiguous. A more rigorous empirical analysis is warranted, such as
> >
> > - comparing the memory and runtime efficiency between the proposed method and standard Transformer in long contexts;
> > - demonstrating the detailed trade-off between task performance (perplexity, retrieval accuracy, etc.) and empirical runtime, especially concerning the swapping-in and -out operations.

---

### Official Review · Reviewer_fLCX · 2023-07-04

**Soundness:** 3 good
**Presentation:** 4 excellent
**Contribution:** 3 good
**Rating:** 5
**Confidence:** 4

**Summary:**

This works proposed a hierarchical structure to organize previous context in blocks and represent them via novel landmark tokens at the end of each block. Additionally, a novel GroupedSofxmax mechanism is proposed to replace the original softmax to enable current token to attend on both local tokens and retrieved previous contexts. Also, a novel positional encoding method is proposed to avoid the positional confusion due to retrieved previous contexts. The author demonstrate the effectiveness of proposed method on several language model datasets.

**Strengths:**

1. Different from the retrieval-based attention on all previous cached contexts in Memorizing Transformer, the proposed method proposes a novel way to split the previous context into blocks and attend on the well-organized blocks based on GroupedSoftmax on the landmark token representation of each token. The whole design is intuitive and feasible.

2. I do appreciate the related work section of this paper. This section discuss almost all feasible solutions on extending context length of Language Models.

**Weaknesses:**

1. My major concern towards the proposed method is that it is too trivial to implement, thus leading to the lower training stability and higher computational timecost. Now the authors store everything like the landmark token representations, previous contextual representations in GPU memory for simple implementation, and thus for each attention layer the stored vectors will take up a lot of GPU memory. The authors talk about the future work regarding storing cached vectors in CPU memory or even disks. However, the computation for the proposed GroupedSoftmax is still deployed on GPU. The major efficiency bottleneck will be the loading and offloading process from GPU to CPU or from CPU to GPU, as this process is several times longer than that of attention computation timecost. Additionally, implementing these cached index on disks seems like impossible as the data transfer timecost from disk to CPU and then to GPU is unacceptable for any real-time applications. Thus, analysis and statistics towards the introduced additional timecost and GPU memory cost will be appreciated.

2. My second concern is that the proposed method introduces too many hyperparameters to control the granularity of the block architecture, including the local context length, the block length, number of blocks, number of retrieved blocks, and attention size. The author should make a constant decision on which group of hyperparameters to use in all experiments or propose a brute-force strategy to select such group of hyperparameters.

3. You only follow Memorizing Transformer to compare the proposed method to Transformer-XL. However, the memorizing transformer, as the most important baseline to your methods, should be also compared with. The memorizing transformer can be regarded as a special case of your model when number of block is 1, local context length is eval length, number of retrieved block is 32, block length is 1 token, attention size is the local context length plus the number of retrieved blocks, and original softmax and attention is applied.

**Questions:**

The presentation of the whole paper is perfect. I think I have understand the details of the paper well thanks to the great presentation. The only issue regarding the paper is that most of references lack of publication journal/conference and the publication year.

---

> ### Author Rebuttal · Authors · 2023-08-08
>
> Dear Reviewer fLCX,
>
> We make the following comments to address your questions and concerns:
>
> 1. We did not observe any training instability due to using landmark attention. Furthermore, as mentioned in the paper, the additional computational costs are negligible especially when combined with Flash Attention. This is because our method facilitates performing inference at any length regardless of the train context length, removing the necessity of a costly training procedure for long context inference. Choosing the alternative approach of training at the target inference length would be much more expensive and scale quadratically with the length during both training and inference.
>
> 2. In Appendix G, we demonstrate how offloading the KV cache to the CPU, along with reduced retrieval flexibility, enables us to extend LLaMA's context length to 32K. As a result, we achieve 98% accuracy in retrieving the pass key. Without our method, performing inference at this context length would be impossible, as the KV cache cannot fit in a single A100 GPU.
>
> 3. The local context length and the attention size are directly determined based on the model’s training context length (which already is a hyperparameter that has to be chosen and depends on the available resources) and the number of retrieved blocks k. These values are provided in Table 1 for simplicity and better comparison. Therefore, the method only introduces two new hyperparameters which are the block size and the number of retrieved blocks. We note that only the block size needs to be chosen at training time and even this can be mitigated by providing the model with blocks of different sizes at training. In early stages of our work, we briefly experimented with dropping some of the landmark tokens, thus merging consecutive blocks, and did not find a noticeable effect on performance. For simplicity we decided to avoid doing this for the final experiments. Overall we do not expect much tuning to be required for the choice of the block size and this value can be chosen based on common sense (keeping the block small enough so that retrieving a single block does not fill the context with unimportant tokens and it can be summarized effectively, but keeping it large enough to use the computational benefits). Since the optimal k can be chosen at inference, it is easy to tune. Our results demonstrate the trade off between using a larger and smaller k. Finally, increasing k beyond a certain point has negligible effect since softmax is applied over the landmarks of the retrieved blocks. For example tokens in the 10-th top block can only have a weight of at most 0.1.
>
> 4. We would like to emphasize that a very important difference between Memorizing transformers and our method is the need to train the model at the large context length which either increases training cost significantly or makes the implementation more complicated since it requires connecting the model to FAISS data structure. The increased training cost makes it challenging to train such a model for large contexts. In contrast, our method allows inference at arbitrary context lengths regardless of the training context length. As a side note, we do not expect the stingy mapping to work for extremely small blocks (such as block size 1) as in that case mapping earlier blocks all to the same position in the stage of finding relevant blocks resembles searching in a bag of words.
>
> We hope that the above replies adequately clarify our contributions and that you would consider raising your score. We would also like to ask that you read the general reply we have provided which addresses common concerns raised by the reviewers. We remain at your disposal to answer any additional questions or comments.

---

> > ### Comment · Reviewer_fLCX · 2023-08-18
> > **Response to Author Rebuttal**
> >
> > Concern 1: I think the description in Line 55-57 leads to the misunderstanding from the reviewers. It is good that you present the benefit for cpu offloading in Appendix G. Therefore, your description in Line 55-57 should be modified and the results in Appendix G should be mentioned here. Meanwhile, some detailed statistics should be presented in a Table, i.e. the timecost with and without cpu offloading for context length = [8k, 16k].
> >
> > Comment 2: Regarding our description "as the KV cache cannot fit in a single A100 GPU", from my understanding, your method replaces the softmax in each decoder layer with designed grouped softmax function with the attention on landmark token and retrieved blocks. If you keep some of the decoder layers as its original architecture and the kv cache of the previous context in these layers does not need to be cached,  in this way you GPU memory requirement can be largely reduced. If you are interested in such implementation, you can verify whether it is good to save the GPU memory cost and simultaneously maintain the performance.
> >
> > Comment 3: You did not mention anything regarding the flash attention in your paper. If you adopt such efficiency technique in your method, please give some details on how you adopt that in your implementation.
> >
> > Comment 4: The author did not convince me that the method demonstrates both the efficiency and capability priority compared with Memorizing Transformer. As for training efficiency, the local context length for memorizing transformer is also 512 which is the same as your method. Memorizing Transformer and your method both introduce some external timecost for attending on previous context. Memorizing Transformer adopts the approximate retrieval with FAISS for acceleration, which brings extra timecost. You method also requires the indexing for previous blocks and reconstruction on the attention matrix. It is hard to demonstrate your method via weak descriptions. I hold the same opinion as reviewer tGXa, and the lack of comparison with Memorizing Transformer keeps this paper a borderline paper. I am ok to accept your paper, but you should consider to reproduce Memorizing Transformer and compare to your method.
> >
> > Comment 5: Towards your explanation "As a side note, we do not expect the stingy mapping to work for extremely small blocks (such as block size 1) as in that case mapping earlier blocks all to the same position in the stage of finding relevant blocks resembles searching in a bag of words.", I believe the previous work like KNN-LM has demonstrated that using token-level retrieval and fusion can boost the perplexity score on language modeling benchmark. Sometimes the token-level retrieval can retrieve the similar token representations which are beneficial for language modeling task, especially when your evaluation metric is token-level perplexity. You should consider to add such ablation study to your Table 2.
> >
> > Therefore, I still keep the final review score as 5.

---

> > > ### Author Response · Authors · 2023-08-20
> > >
> > > Dear Reviewer,
> > >
> > > Thank you very much for participating in the rebuttal and providing additional comments. We hope the following clarifications addresses your concerns further.
> > >
> > > 1. Thank you for your suggestion. We will update the manuscript as suggested to refer the reader to Appendix G.
> > >
> > > 2. When generating auto-regressively, the KV cache is needed in the standard Transformer architecture as well, i.e. even when all the layers are the original ones. In other words the need for the KV cache **is not** a side-effect of our method. The need for KV cache arises to avoid the need to recompute all the intermediate vectors for previous tokens when the next token is fed through the transformer. In comparison to the original architecture, our method reduces the computation by a factor of block size and alleviates memory requirements by allowing off-loading the KV cache. The only increase in memory requirements when using our method comes from the additional landmark tokens which is negligible.
> > >
> > > 3. None of results included in the paper are based on Flash Attention. However, we discuss the possibility of combining our method with Flash Attention in Appendix F. Flash Attention computes the output of attention by processing the tokens in blocks. By using the same block size for both flash attention and landmarks, it is possible to implement a fused version of landmark attention. Since the submission, we have explored this possibility and implemented the fused version allowing us to obtain better performance and reduced memory footprint. However, for this paper, we use the high level implementation.
> > >
> > > 4. We want to emphasize that the exact difference is that our method **does not** incur an additional time cost for retrieval **at training** whereas Memorizing Transformer does as you described. The mentioned retrieval and indexing process is only applied at inference.
> > >
> > > 5. During our experiments, we found out that it is necessary to have the correct indices at least for the last few blocks as we discussed in lines 242-247. As a guess, the success of previous methods can be possibly because these methods include the retrieval during the training, allowing the model to adapt to some extent. However, verifying this falls outside the scope of this work.
> > >
> > > We hope that the above replies adequately clarify our contributions and that you would consider raising your score. We remain at your disposal to answer any additional questions or comments.

---

### Official Review · Reviewer_Hd8A · 2023-07-19

**Soundness:** 2 fair
**Presentation:** 3 good
**Contribution:** 3 good
**Rating:** 5
**Confidence:** 5

**Summary:**

The paper proposes a new attention mechanism that uses "landmark" tokens to allow access to long contexts while retaining the flexibility of standard attention. The landmark token represents each block of the input context, and the attention mechanism is trained to use landmark tokens to select relevant blocks. This allows retrieving blocks directly through attention instead of separate retrieval mechanisms. Experiments including language modeling and fine-tuning LLaMA 7B show landmark tokens' effectiveness.

**Strengths:**

- The landmark design is interesting, which maintains the random-access flexibility of standard attention, unlike other retrieve-based approaches. It enables the possibility of processing arbitrarily long contexts and practically extends the context length of large LMs like LLaMA by 5x.
- This method is able to reduce memory footprint since only landmark tokens need to be stored in GPU and some KV cache can be offloaded to CPU. Besides, the landmark approach needs fewer retrieved tokens per step, compared with Transformer-XL.


**Weaknesses:**

- One of the advantages of the landmark approach in the paper, is to reduce memory usage by swapping out (to CPU memory or to disk) all regular tokens’ cached key-value vectors (line205). However, this is not tested in experiments, as stated in line 56:
>For simplicity, we focus our experiments on storing everything in GPU memory, but we note that the above techniques can be directly applied in large-scale settings.
It is better to add the related experiments about memory saving to validate the effectiveness of this approach.
- The language modeling experiment is not that persuasive. The perplexity of the landmark approach is not better than Transformer-XL, although it uses a smaller attention size. It is still not sure whether the language modeling ability as well as the memory consumption is supreme to transformer-XL.
- The experiment of finetuning LLaMA 7B is not that persuasive, LLaMA-2k is not good at extrapolation, and this baseline is really weak. Besides: i) only test Retrieval tasks, lack of other common long document understanding/generation tasks (e.g. scrolls benchmark or many-shot in-context learning); ii) there are many other approaches targeting long document processing (listed in Related work), lack of experiments to compare some of these baselines.

**Questions:**

- In Table 2, for different levels of retrieval flexibility, the change of perplexity is small. Does this mean the choice of $k$ is not that important?
- What is the effect of doing memory-offloading for unused KV cache? To what extent it can save memory consumption?
- Can you explain and compare the choice of pre-training the landmark LM from scratch and finetuning a landmark LM based on the existing LM like LLaMA.
- The line spacing between line 339-340, and line 347-348 is too narrow.


**Limitations:**

The authors have not addressed their limitations officially in the conclusion but admitted the computation limitations in the middle of the paper and also stated their future work.

---

> ### Author Rebuttal · Authors · 2023-08-08
>
> Dear Reviewer Hd8A,
>
> We make the following comments to address your questions and concerns:
>
> 1. In Appendix G, we demonstrate how offloading the KV cache to the CPU, along with reduced retrieval flexibility, enables us to extend LLaMA's context length to 32K. As a result, we achieve 98% accuracy in retrieving the pass key. Without our method, performing inference at such long context lengths would be impossible, as the KV cache grows and cannot fit in a single A100 80GB GPU.
>
> 2. We present Transformer XL as a baseline to show that landmark attention can perform comparably in utilizing long contexts. However, despite performing slightly better, Transformer XL has inherent limitations as it cannot directly access earlier tokens, hindering its ability to perform certain tasks, such as retrieval. In contrast, our LLaMA experiments demonstrate that Landmark Attention successfully retrieves and attends to early tokens.
>
> 3. As you mentioned the original LLaMA 7B was trained on 2k tokens and can only perform inference up to this length. We want to demonstrate that fine-tuning using landmark attention (at 512 context length) makes the model capable of handling arbitrarily long context lengths. Our experiments clearly verify this for 32k tokens. Please note that all existing methods for handling long contexts usually require training at the target inference length which increases the training cost or sacrifices the ability to directly attend to any tokens.
>
> 4. We point out that most existing methods for retrieval use a separate embedding model to perform the retrieval which needs to be trained separately, increasing the training cost. In contrast, our method allows using the model itself to perform the retrieval. Furthermore, many of the proposed retrieval-based methods usually focus on retrieving whole documents rather than finding relevant parts of the input. Extending these methods to be capable of identifying small relevant blocks of the input is beyond the scope of our work and requires further research. Other existing methods for augmenting transformers with memory or long context capabilities either remove the capability to directly attend to each token , which is unlikely to perform well for retrieving an exact pass key, or involve a costly training method which is challenging to perform for a model as large as LLaMA. We have also left further evaluation of the fine-tuned model on other tasks such as summarization as future work given the resource limitation.
>
> 5. When the set of retrieved blocks is allowed to change for each token and each head, increasing k beyond certain values can have a negligible effect since softmax is applied over the landmarks of the retrieved blocks. For example tokens in the 10-th top block can only have a weight of at most 0.1. However, in a less flexible regime where the retrieval set remains the same across all tokens (but can vary across heads), k can have a more noticeable effect. For example, as shown in Table 2, increasing k from 2 to 4 in this setting improves performance from 15.48 to 15.10.
>
> We hope that the above replies adequately clarify our contributions and that you would consider raising your score. We would also like to ask that you read the general reply we have provided which addresses common concerns raised by the reviewers. We remain at your disposal to answer any additional questions or comments.

---

> > ### Comment · Reviewer_Hd8A · 2023-08-18
> > **Thank you for clarification**
> >
> > Thank you for the clarifications which makes the paper clearer. However, some concerns remain unsolving, such as: lack of testing in long document understanding/generation tasks (zero/few-shot instead of fine-tune) and the comparison between the pre-training landmark LM from scratch and finetuning a landmark LM based on LLaMA.

---

### Official Review · Reviewer_DxW4 · 2023-08-03

**Soundness:** 4 excellent
**Presentation:** 3 good
**Contribution:** 4 excellent
**Rating:** 8
**Confidence:** 5

**Summary:**

The authors propose a new transformer architecture, the Landmark Transformer, which uses a novel approach to the self-attention mechanism to allow the model to handle longer sequences.

The Landmark Transformer introduces a new type of token called a "landmark token" which acts as a gateway to a block of tokens. The blocks are assigned to these landmark tokens in a way that allows the model to handle longer sequences while maintaining computational efficiency. The landmark tokens are distributed across the sequence and form a part of the model's input. They are processed in a particular order and are used to control the flow of attention in the transformer model. In some ways, this could also be viewed as introducing hierarchy (and or tree-structures) into the context window. They provide a higher level abstraction over local sets of tokens.

The authors propose a "Grouped Softmax" operation, which is used to calculate attention scores in the self-attention mechanism. This operation allows the transformer to regulate its attention across different blocks of tokens. The Grouped Softmax operation is applied to both the query and key vectors in the self-attention mechanism. The authors also propose an efficient method for calculating these Grouped Softmax operations.

The authors evaluate their proposed models on the PG-19 dataset and an arXiv math dataset. The results demonstrate that their models can handle longer sequences while maintaining good performance when compared to the baseline .

**Strengths:**

The paper offers a novel and simpple approach to extending the context length of transformer models.

The multi-resolution nature of this approach allows for exact retention of fine-grained details and is very amenable to model interpertability approaches.

This method can be applied post-hoc, is extensible to tools/frameworks that can better take advantage of CPU processing and system RAM.

This approach can be used to directly build document retrievers without any further training required.

The approach is also quite efficient, requiring at most 4 A100 GPUs to achieve the results presented in this work.

This approach has the potential to enable heirarchical scaling of the context window, which when combined with an external cache could enable quadratic/exponential growth of the context window.

The illustrations in the article help alot in conveying the core ideas.

Overall, a solid, timely and well-justified piece of work.

**Weaknesses:**

There should be more diversity in the models and datasets benchmarked against in this paper.


Sections 3.1 (presenting grouped softmax) and the middle paragraph os section 3.2 are confusing and less clear than the rest of the article.

**Questions:**

There is a typo on line 185: exmaple -> example

where these 80GB or 40GB version of the a100?

could you provide a graph of how memory/compute/time scales with block size and length?

Experiments to demonstrate the memory offloading capabilities of this approach would improve this work.

It would alse be good to benchmark against the memorizing transformer and similar competing appraoches.

I will improve my score if these questions are answered and the weaknesses raised are addressed.

**Limitations:**

There is no discussion of the social impact of this work, it should be added though.

---

> ### Author Rebuttal · Authors · 2023-08-08
>
> Dear Reviewer DxW4,
>
> We make the following comments to address your questions and concerns:
>
> 1. The language modeling experiments were done using 4 A100 40GB GPUs. LLaMA fine-tuning was done using 8 A100 80GB GPUs.
>
> 2. We have successfully used offloading parts of KV cache to CPU to perform inference at 32k context lengths using LLaMA. We have described this result in Appendix G.
>
> 3. Here is an overview of the effects of increasing block size or context length during training and inference.
>     * During training, the block size has almost no effect on time or memory usage since we keep the context size (including the landmarks) fixed. Since we use the standard training procedure for transformers, computation time scales quadratically with the context size (during training). We note that, a future work can look into using a similar retrieval method used at inference during training to reduce the training computation by block size as well.
>     * During inference, when using landmark attention, landmark tokens need to be also stored in memory which slightly increases the memory usage and compute time by a factor of 1/(block size). Thus increasing block size reduces memory usage while also reducing the time needed to find relevant blocks since there are less blocks. However, if the number of retrieved blocks $k$ is kept constant, the length of the chunks have to become smaller so the longer retrieved blocks and the current chunk can fit into the model's maximum allowed context length, i.e. the train context length. Therefore, increasing the block size increases the number of chunks the input has to be broken into, which slows down the inference. This trade-off can be seen in the Figure 1 of the PDF attached to our global rebuttal, showing the inference time for different block sizes and different values of $k$.
>     * When increasing the context length, the standard transformer’s memory usage and operation time grows quadratically since it needs to compute the full attention matrix. When using landmark attention, the attention matrix is only computed for a fixed number of blocks, which means the total memory usage and operation time for computing the attention matrix increases only linearly. However the bottleneck becomes finding the set of relevant blocks. As we mentioned it is possible to use a kNN data structure to do this operation efficiently as well but we leave exploration of such implementation for future work. With the direct (not kNN) implementation we used for our experiments, the memory usage and operation time of this operation is also quadratic in context length but is reduced with the noticeable factor of block size (e.g. 50x reduction). The implementation still needs to be done efficiently for this improvement to be observable. Finally, we note that using flash attention, it is possible to reduce the quadratic memory usage for both our method and standard transformer to linear. We have already implemented this version of our method in Triton and have seen its success. Please note that the reduction of operation time by the noticeable block size factor remains even with flash attention.
>
> 4. This unique characteristic of our method sets it apart from previous approaches like Memorizing Transformers, which require expensive training with large context lengths. As a baseline, we implemented Transformer-XL to showcase that we can achieve comparable performance in utilizing long contexts. Comparing against additional baselines becomes a challenging task considering the increased training costs and the complexity of implementing such methods (e.g., Memorizing Transformers rely on FAISS during training).
>
>
> We hope that the above replies adequately clarify our contributions and that you would consider raising your score. We would also like to ask that you read the general reply we have provided which addresses common concerns raised by the reviewers. We remain at your disposal to answer any additional questions or comments.

---

### Author Rebuttal · Authors · 2023-08-08

Dear Reviewers,

We would like to address some of the common concerns raised by the reviewers with the following comments:

* We have successfully used offloading parts of KV cache to CPU to perform inference at 32k context lengths using LLaMA. We have described this result in Appendix G. Note that while offloading to CPU might make the inference a bit slower, without our method performing inference at such long context lengths becomes impossible on a single A100 80GB GPU since it is not possible to store the KV cache.

* One of the major advantages of Landmark Attention is its ability to decouple the training and inference context lengths. This means that the model can be trained at a limited context length, thus avoiding increased training costs. The trained model can then be effectively used for making inferences at any context length. We have extensively demonstrated this feature in our language modeling experiments where we train our models at context length 512 and successfully infer at much larger context lengths, such as 4096. Similarly, with LLaMA, we fine-tune at context length 512 using a model originally trained at 2k context length, and we can perform inference at a much larger 32k context length with excellent results.

* This unique characteristic of our method sets it apart from previous approaches like Memorizing Transformers, which require expensive training with large context lengths. As a baseline, we implemented Transformer-XL to showcase that we can achieve comparable performance in utilizing long contexts. Comparing against additional baselines becomes a challenging task considering the increased training costs and the complexity of implementing such methods (e.g., Memorizing Transformers rely on FAISS during training).

* We note that while our method performs comparably (but not better than) Transformer-XL it has the advantage of being able to directly attend to all previous tokens. This is important to utilize the full power of attention, for example allowing our method to excel at retrieving tasks as demonstrated in our LLaMA experiment.

In addition to these general comments, we have provided individual responses to the reviewers to address their specific concerns. We hope that these explanations further clarify the contributions of our work. We are available to address any further comments or questions you may have.

---

### Decision · Program_Chairs · 2023-09-21

**Decision:**

Accept (poster)

**Comment:**

Meta Review for Random-Access Infinite Context Length for Transformers

As reviewer DxW4 had summarized, this work proposes a new transformer architecture, the Landmark Transformer, which uses a novel approach to the self-attention mechanism to allow the model to handle longer sequences. The Landmark Transformer introduces a new type of token called a "landmark token" which acts as a gateway to a block of tokens. The blocks are assigned to these landmark tokens in a way that allows the model to handle longer sequences while maintaining computational efficiency.

Several innovations had been proposed, including the grouped softmax operation and its efficient calculation. This operation allows the transformer to regulate its attention across different blocks of tokens. The Grouped Softmax operation is applied to both the query and key vectors in the self-attention mechanism.

This work clearly adds value to the community, especially in the context of improving generative AI. There are no clear negative issues or weaknesses. Looking at the feedback, 4 reviewers gave borderline accept (5), and one high confidence review gave it a strong accept (8). The reviewer's feedback has been incorporated into the manuscript, and I believe it will be a paper that will offer good value and interest to the NeurIPS community.